# A Bayesian Approach to Data Point Selection

**Xinnuo Xu**[†][*]
Microsoft Research Cambridge
xinnuoxu@microsoft.com

**Minyoung Kim**[†]
Samsung AI Center Cambridge, UK
mikim21@gmail.com

**Royson Lee**
Samsung AI Center Cambridge, UK
royson.lee@samsung.com

**Brais Martinez**
Samsung AI Center Cambridge, UK
brais.mart@samsung.com

**Timothy Hospedales**
Samsung AI Center Cambridge, UK
University of Edinburgh, UK
t.hospedales@ed.ac.uk

## Abstract

Data point selection (DPS) is becoming a critical topic in deep learning due to the ease of acquiring uncurated training data compared to the difficulty of obtaining curated or processed data. Existing approaches to DPS are predominantly based on a bi-level optimisation (BLO) formulation, which is demanding in terms of memory and computation, and exhibits some theoretical defects regarding minibatches. Thus, we propose a novel Bayesian approach to DPS. We view the DPS problem as posterior inference in a novel Bayesian model where the posterior distributions of the instance-wise weights and the main neural network parameters are inferred under a reasonable prior and likelihood model. We employ stochastic gradient Langevin MCMC sampling to learn the main network and instance-wise weights jointly, ensuring convergence even with minibatches. Our update equation is comparable to the widely used SGD and much more efficient than existing BLO-based methods. Through controlled experiments in both the vision and language domains, we present the proof-of-concept. Additionally, we demonstrate that our method scales effectively to large language models and facilitates automated per-task optimization for instruction fine-tuning datasets.

## 1 Introduction

Practical machine learning efficacy is heavily dependent on the choice, quality and quantity of training data, especially so in the case of neural networks that can easily fit every detail of the training set. This leads to challenges from how to learn reliably with imbalanced data [22], noisy data, noisy labels [46], and so on. Similarly there is often a key subset of data, which is most informative for a given learning problem, but buried among a much larger set of less relevant data. If the most salient data could be efficiently identified, learning could potentially be accelerated [15]. All these challenges are only growing in the era of large scale training on web-scraped data, where curation and gold-standard quality control are not feasible.

Data Point Selection (DPS) algorithms aim to address these challenges by filtering or re-weighting the training data to reduce noise, imbalance, irrelevant background data and so on. The most established

---

[*]Part of this work was completed while Xinnuo was affiliated with Samsung AI Center Cambridge, UK.
[†]Equal Contribution.

38th Conference on Neural Information Processing Systems (NeurIPS 2024).

family of approaches [19, 15, 41, 45, 62] to DPS falls under the bi-level optimization or meta-learning umbrella, where one wraps the conventional learning problem with an outer loop that optimizes the dataset itself, so as to maximise performance on some validation set. These methods vary in the choice of their outer optimization variable (e.g., data point weights [41, 45, 19] vs mini-batch sampler [15]), the method of computing meta-gradients (e.g., reverse mode differentiation [45, 62] or reinforcement learning [15]), and the customization of their losses and other design parameters for the different scenarios (e.g., label-noise [45, 41], etc). However, all the BLO approaches are quite expensive in computation and/or memory, which limits their applicability to the most salient use case of large models trained on large web data.

In this paper we revisit the DPS problem from the perspective of Bayesian learning. Rather than constructing expensive nested optimization problems whose convergence is hard to analyse, we treat it as a problem of inferring the joint posterior over the main neural network parameters and instance-wise weights induced by a second weight-estimation neural network. This framework has several advantages in terms of being more efficient and scalable than typical BLO competitors and having a clear convergence guarantee. It is also able to address a variety of DPS-related problems – from noise and imbalance to data curation – within a single framework.

Our empirical results present proof of concepts for all these capabilities on a variety of learning tasks in vision and language. We also show a use case of automating Large Language Models (LLMs) instruction fine-tuning (IFT) data curation for specific downstream tasks. The available IFT datasets are large, diverse, and of varying quality. This means that a key activity for natural language processing (NLP) researchers and developers is often finding the right composition of IFT data sources to optimize particular use cases. Our framework can automatically resample and curate the wide array of available auxiliary IFT data to optimize performance for each NLP task of interest. To our knowledge, no BLO alternative has been demonstrated on billion-parameter scale LLMs. We name our approach **BADS** (**Ba**yesian **D**ata Point **S**election).[3]

## 2   Our Approach

### 2.1   Problem Setup

For the DPS problem, we assume that we are given two datasets: the *train* set $\mathcal{D}_t = \{z_i^t\}_{i=1}^{N_t}$ and the *meta* set $\mathcal{D}_m = \{z_i^m\}_{i=1}^{N_m}$. Each data point $z_i$ can be an input-target pair $z_i = (x_i, y_i)$ in the traditional (class-)labeled data scenarios. But in the autoregressive generative model scenarios (e.g., LLM), $z_i$ is simply a sequence of tokens in which the inputs and targets are rather implicitly defined (e.g., all the tokens up until current time as input and the next token as target). We will use the notation $l(z_i; \theta)$ for the loss of the model $\theta$ on the data point $z_i$, which must be well-defined in both scenarios.

The meta dataset $\mathcal{D}_m$ is considered as *in-domain*, meaning that the distribution of $\mathcal{D}_m$ matches that of the downstream test task of interest. The size of $\mathcal{D}_m$, denoted by $N_m$, is typically small, due to the cost of curation/annotation processes in practice. The train dataset $\mathcal{D}_t$ consists of *out-of-domain* samples, possibly noisy, imbalanced, and uncurated, but the size $N_t$ is usually large. The goal of DPS is to select a (soft weighted) subset of the train set $\mathcal{D}_t$ with the guidance of the meta set $\mathcal{D}_m$, so that the model trained on the selected train and meta dataset points performs well. Typical baselines include training with the meta-set alone, or union of meta and train sets.

Perhaps one of the most widely adopted DPS techniques is the bi-level optimisation (BLO) formulation of the problem [41, 58]. Letting $w_i$ ($\geq 0$) be the weight (or importance) variable associated with the training data point $z_i^t$, the main intuition is to find the weights $w \in \mathbb{R}_+^{N_t}$ such that the model $\theta$ trained with the weighted train data with weights $w$ yields the best performance on the meta set. More formally, this leads to the following BLO problem:

$$\min_{w \in \mathbb{R}_+^{N_t}} \sum_{j=1}^{N_m} l(z_j^m; \theta^*(w)) \quad \text{s.t.} \quad \theta^*(w) = \arg\min_\theta \sum_{i=1}^{N_t} w_i \cdot l(z_i^t; \theta). \tag{1}$$

However, a critical drawback is that solving this difficult problem is costly in computation and/or memory, and unrealible due the practical heuristics required. Typical BLO solutions to obtain the

---

[3]The code for this paper is available at `https://github.com/XinnuoXu/BADS`.

hypergradient $dl/d\omega$ rely on approximate Hessian estimation or reverse mode differentiation with few-step SGD approximation of the inner optimisation. Aside from cost, for practical neural network implementations computed over *minibatches*, there is no theoretical guarantee for convergence.

## 2.2 (Our Approach) Bayesian Data Point Selection (BADS)

We view the DPS problem from a completely different perspective, and tackle it via Bayesian learning. Our model's generative process, that is, the graphical model representation, is depicted in Fig. 1. The main neural network model parameters set $\theta$ is a random variable, which can generate data points in the meta dataset $\mathcal{D}_m$ (precisely speaking, the backbone $\theta$ generates the target part of each data point). To make use of the train set $\mathcal{D}_t$ in an appropriate way, we constrain $\theta$ to follow a prior distribution governed by the weighted train data with weights $w \in \mathbb{R}_+^{N_t}$ which are also random variables. Before observing the meta set $\mathcal{D}_m$, the weight vector $w$ follows a

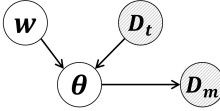

Figure 1: Graphical model for *BADS*. Shaded nodes, representing curated ($D_m$) and uncurated ($D_t$) data, are evidence. Unshaded nodes, including model $\theta$ and instance weights $w$, are random variables.

prior distribution $p(w)$ – a specific distributional choice for $p(w)$ will be discussed later. Given $w$ and $\mathcal{D}_t$, our backbone $\theta$ has to be compatible with the weight data $\{(w_i, z_i^t)\}_{i=1}^{N_t}$. This can be interpreted as placing a *weighted-data-driven prior* on $\theta$, more specifically,

$$\text{(Weighted-data-driven prior)} \quad p(\theta|w, \mathcal{D}_t) \propto p(\theta) \cdot \prod_{i=1}^{N_t} p(w_i, z_i^t|\theta) \tag{2}$$

where $p(\theta)$ is a base prior (e.g., 0-centered Gaussian that amounts to weight decay regularisation), and $p(w_i, z_i^t|\theta)$ can be defined from the loss, e.g., $\exp(-w_i \cdot l(z_i^t; \theta))$, following the conventional tricks [36, 25]. Then given $\theta$, the meta data are generated following the likelihood defined as:

$$\text{(Likelihood)} \quad p(\mathcal{D}_m|\theta) \propto \prod_{j=1}^{N_m} \exp(-l(z_j^m; \theta)) \tag{3}$$

The equations (2) and (3) fully constitute the prior and likelihood for our Bayesian model. Our ultimate goal is to describe the distributions of $\theta$ and $w$ *after* observing all evidences $\mathcal{D}_t$ and $\mathcal{D}_m$, which boils down to the posterior inference $p(\theta, w|\mathcal{D}_t, \mathcal{D}_m)$. Formally, we have:

$$\text{(Posterior)} \quad p(\theta, w|\mathcal{D}_t, \mathcal{D}_m) \propto p(w) \cdot p(\theta|w, \mathcal{D}_t) \cdot p(\mathcal{D}_m|\theta) \tag{4}$$

The detailed derivations for Eq. (4) can be found in Appendix A. However, it is widely known that (4) does not admit any closed-form expressions. One main difficulty arises from the intractable normalizing constant in (4).

**Stochastic Gradient Langevin Dynamic Sampling.** For computationally efficient posterior inference, we adopt the stochastic-gradient MCMC technique, specifically the Stochastic Gradient Langevin Dynamic (SGLD) sampling [55]. Applied to our model, we can obtain samples from the posterior $p(\theta, w|\mathcal{D}_t, \mathcal{D}_m)$ by running the Langevin dynamic system (until convergence, i.e., mixing):

$$[\theta, w] \leftarrow [\theta, w] + \frac{\eta}{2} \nabla_{\theta, w} \log p(\theta, w|\mathcal{D}_t, \mathcal{D}_m) + \epsilon\sqrt{\eta}, \quad \epsilon \sim \mathcal{N}(0, I) \tag{5}$$

where $\eta$ is a small (constant) step size. There are two critical benefits: i) Since we differentiate the log-posterior, the difficult normalizing constant in (5) will disappear; ii) The update (5) is essentially gradient descent with additive Gaussian noise, leading to a computationally efficient update.

Going one step further, even though the log-posterior involves the entire train data (and entire meta data), it is shown in [55] that the stochastic-gradient version (SGLD) that replaces the whole batch likelihood with a minibatched one, theoretically guarantees that the SGLD update converges to the posterior samples. More specifically, the SGLD update equations (one for $\theta$ and the other for $w$) can be written as follows:

$$\theta \leftarrow \theta + \frac{\eta}{2} \nabla_\theta \Big( \log p(\theta) - N_t \cdot \mathbb{E}_{i \sim \mathcal{B}_t}\big[w_i \cdot l(z_i^t; \theta)\big] - N_m \cdot \mathbb{E}_{j \sim \mathcal{B}_m}\big[l(z_j^m; \theta)\big] \Big) + \epsilon_\theta\sqrt{\eta} \tag{6}$$

$$w \leftarrow w + \frac{\eta}{2} \nabla_w \Big( \log p(w) - N_t \cdot \mathbb{E}_{i \sim \mathcal{B}_t}\big[w_i \cdot l(z_i^t; \theta)\big] \Big) + \epsilon_w\sqrt{\eta} \tag{7}$$

where $\mathcal{B}_t$ and $\mathcal{B}_m$ are minibatches from $\mathcal{D}_t$ and $\mathcal{D}_m$, respectively, and $\epsilon_\theta, \epsilon_w \sim \mathcal{N}(0, I)$ are independent Gaussian samples.

Repeating (6) and (7) for a sufficient amount of iterations (until we reach good mixing) leads us to posterior samples $(\theta, w)$. There are several options to take these samples for a final model for test prediction. One option is to collect latest $M$ samples (either consecutive collection or thinning to take every $k$th samples) from the iterations, and either take the average as *posterior means* or perform full Bayesian treatment with the collected samples. Alternatively, we can just take the last single iterate $(\theta, w)$ as a point representative for the posterior distribution. For simplicity, we take the latter approach, which also works well empirically.

## 2.3 Interpretation and Benefits

**Interpretation**  We discuss several intuitions and implications of our proposed approach (Eq. 6-7). First, looking at the $\theta$ update Eq. (6), our model essentially updates $\theta$ in a way that it decreases the loss on the *combined data* of the whole meta data points and the weighted train data points with the current weights. This is a fairly intuitive strategy provided that the weights are properly determined. Then the next question is how the weights are determined. If we inspect the $w$ update (Eq. 7), and take the gradient of the expected loss term with respect to $w$ directly, we see that: i) those train data points $z_i^t$s with *smaller* losses at current backbone $\theta$ will get *higher* weights $w_i$s; ii) those train data points $z_i^t$s with *larger* losses at current backbone $\theta$ will get *lower* weights $w_i$s. This essentially means that our model performs **loss alignment** for DPS – In the course of training/update, once the backbone $\theta$ enters a good regime in the parameter space such that $\theta$ can assign (valid) low loss values on the in-domain meta data points, then it starts putting high weights on those train data points that have low losses under the current backbone. In other words, the model will assign high weights to those train data points that are well-aligned with the meta data points in terms of loss.

**Benefits over BLO**  Our Bayesian approach provides several benefits over BLO: (1) Efficiency. Our SGLD is efficient so does not require computationally demanding Hessian computations like implicit function theorem based methods (cf: [19]) or huge memory demand like reverse-mode differentiation methods [41, 19]. (2) Sparsity. Our method straightforwardly achieves sparsity on the $w$ weights allowing efficient sample selection unlike [19]. (3) Reliability. BLO-based methods rely on approximations (truncation, or Hessian approximations) for practical feasibility that make finding optimal solutions unreliable. Our straightforward Bayesian approach has reliable convergence properties thanks to being a standard application of SGLD.

**Convergence of our SGLD algorithm**  In Appendix C we provide a theorem showing that our SGLD algorithm converges to the true posterior. Our analysis is based on [64] where we make some adjustments for our case.

## 2.4 Implementation Details

**Choice of Priors**  For the base prior $p(\theta)$, i.e., the prior before being driven by the weighted data, we adopt 0-mean Gaussian, which amounts to adding the weight decay regularisation for $\theta$. For the weight prior $p(w)$, we have made a careful design effort to come up with a viable sparsity inducing prior. Although encouraging sparsity in learned weights is ideal to avoid overfitting, during our initial experiments we have found that most of the weights eventually tend to vanish to 0, which is not what we actually want. We need to be able to impose both sparsity and a certain level of non-zero weights. To this end, we first introduce a hyperparameter $\beta$ (e.g., 0.01) for the target sparsity level that we want to attain. Roughly saying, among the $N_t$ training data points, we aim to select $\lfloor N_t \cdot \beta \rfloor$. For the (soft) weights, we impose $\sum_{i \in \mathcal{D}_t} w_i \approx \lfloor N_t \cdot \beta \rfloor$, which can be encoded in the prior form as:

$$p(w) \propto e^{-\left( \sum_i w_i - \lfloor N_t \cdot \beta \rfloor \right)^2 / 2\sigma^2} \tag{8}$$

where $\sigma$ controls the strength of the regularisation. One technical difficulty in directly plugging (8) into the weight update (7) is that we have to load the whole $\{w_i\}_{i=1}^{N_t}$ in memory for backprop. To avoid this issue, we use the following fact:

$$\sum_{i \in \mathcal{D}_t} w_i \approx \sum_{i \in \mathcal{B}_t} w_i + (N_t - |\mathcal{B}_t|) \cdot \bar{w} \tag{9}$$

where $\bar{w}$ represent historic running average of the entire weights. We basically build a computation graph only for the first term of batch $\mathcal{B}_t$ weight sum, and regard the (historic) running average of the entire weights as constant during backprop. After each SGLD iteration, we update the running weight average with the new updated weights on the recent batch. We use the simple averaging scheme for the running average. To approximate the average weight $\bar{w}$ precisely, we only conduct the average over the most recent $s_{avg}$ step.

**Introducing impact constants**   In the SGLD principle, we have the log-likelihood terms that are proportional to the sizes of the datasets. In particular, we have $N_t$ and $N_m$ in (6). However, this scheme does not properly capture our preference to the in-domain meta data set in contrast to the noisy, out-of-domain train data set. To this end, we introduce the impact constants (hyperparameters) in the update equations where we downweigh or upweigh the loss terms of train and meta sets.

**Weight Network**   Instead of directly optimising individual weights $w_i$, we can consider a weight network, $w_i = w(z_i^t; \phi)$, a neural network with parameters $\phi$ that takes the train data point $z_i^t$ as input and returns its weight $w_i$ as output. We can then regard $\phi$ as random variables and the $w$ update equation can be modified accordingly for $\phi$ update straightforwardly. This weight network approach can be useful for smoothing/regularising the output weights thanks to the smooth functional property of neural networks. Furthermore, if one needs to supplement the train dataset with extra new train samples after the model training, the learned weight network can be used for assigning weights or selecting samples from the new set, without retraining the whole model from the scratch. The posterior distribution similar to Eq. (4) is derived in full detail in Appendix B.

# 3   Experiments: Proof of Concept

In this section, we assess the effectiveness of the proposed *BADS* method in three critical scenarios where DPS is essential: Data Balancing, Data Denoising, and Efficient Learning. We begin by introducing the baseline systems and then present the experimental results for each of these scenarios.

## 3.1   Baselines

There are three types of DPS setups with different supervision signals:

- *Unsupervised DPS* selects data without the guidance of a held-out meta set [47, 42]. Instead, it is guided by human-defined hypotheses, such as "challenging examples improve model performance". This approach aligns with curriculum learning. We include the online variant of AskLLM [42], i.e. **AskLLM-O**, in our baseline comparisons. It selects examples from training set by querying a pretrained OpenLLaMA 3B to obtain the sampling score for each training sample.[4]

- *Self-supervised DPS* selects data with the guidance of a held-out meta set. However, the meta set does not share the same data distribution as the targeted test set [13, 4, 14, 49]. Typically, the examples in the meta set are selected from the training set based on specific hypotheses, such as "learnable examples enhance model performance". We include two approaches in our baseline comparisons: **Contrastive Data Selection (CDS)** [49] is tailored for data denoising. The algorithm assigns weights to each data point in $\mathcal{D}_t$ according to the difference between the *denoised* and the *noisy* log probability, predicted using a denoised and a noisy model trained on a clean dataset and an uncurated dataset, respectively. These weights are then used to sample data points from minibatches in the training of LM.[5] Similar to CDS, **ClassAct** utilizes small proxy models trained on a limited portion of $\mathcal{D}_t$ to calculate learnability scores for the remaining training data points.[6]

- *Meta-set guided DPS* selects data with the guidance of a small meta set that shares the same distribution as the test set, aiming to train a model that excels specifically on the target test set. The test set may encompass one or multiple downstream domains or tasks. This DPS is closely related to meta learning, domain adaptation, and transfer learning. Current methods primarily rely on **Bilevel Optimization (BLO)** for purposes such as data denoising [19, 37], data balancing [41],

---

[4]Since [47] showed that unsupervised DPS can amplify class imbalances, and open-source LLMs generally do not accommodate vision input, we compare to AskLLM-O only in the LLM fine-tuning use case in Section 4.

[5]In our experiments, we use *Mixing* and *Meta_only* as the noisy and denoised model.

[6]To fairly compare the selection mechanism, we replaced their meta set using our meta set.

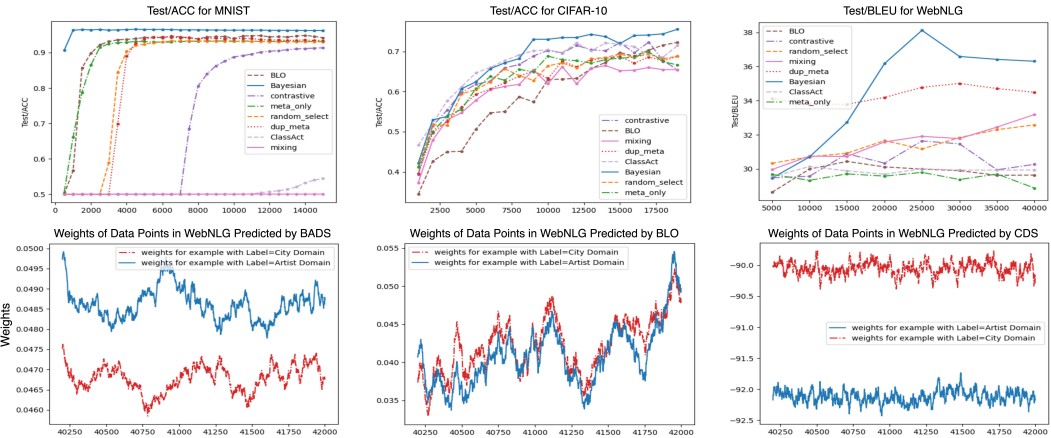

Figure 2: Proof-of-Concept experiment results. The **top** row displays the overall test performance across the three scenarios throughout the training phase, with x and y axis denote the training steps and the evaluation metrics, respectively. The **bottom** row visualizes the model-predicted weights of data points in each mini-batches in the final 2000 steps in WebNLG training (scenario 3). x and y axis show the training steps and average weights, respectively. Data points in **blue** color are expected to get higher weights compared to their counterparts (in **red** color).

and efficient learning [28, 59, 29]. Considering both performance and code availability, we use the online BLO[7] [41, 19] as our baseline. Our approach, **BADS**, also falls under this category.

To ensure a fair comparison, we also introduce several baselines that train the backbone models using different combinations of the meta set and training set: **Mixing** trains the model using a combination of the *train* set $\mathcal{D}_t$ and the *meta* set $\mathcal{D}_m$. **Meta_Only** trains the model exclusively on $\mathcal{D}_m$. **Random_Select** uses $\mathcal{D}_m$ combined with a randomly selected subset from $\mathcal{D}_t$. **Duplicate_Meta** utilize $\mathcal{D}_t$ along with multiple copies of the *meta* set, duplicating $\mathcal{D}_m$ until it matches the size of $\mathcal{D}_t$.

Note that, the selection ratio/sparsity level in *AskLLM-O*, *ClassAct*, *Random_Select*, *BLO*, and *CDS* is the same as in *BADS*.[8]

## 3.2 Scenario 1: Data Balancing (MNIST)

In this scenario, we assess the model's capability to manage an imbalanced *train* set, $\mathcal{D}_t$. Despite being trained on this imbalanced dataset, the models are expected to perform effectively on a balanced test set. Following the setup in [41], we use the standard MNIST handwritten digit classification dataset [33] to create a class-imbalanced binary classification task. A total of 5,000 images from classes 4 and 9 were selected as the *train* set $\mathcal{D}_t$, with class 9 dominating the training data distribution (4,975 examples) and class 4 having only 25 examples. A balanced *meta* set $\mathcal{D}_m$ is created by selecting another 25 examples from each of these two classes, ensuring no overlap between $\mathcal{D}_t$ and $\mathcal{D}_m$. The models are tested on the original MNIST test set, including only classes 4 and 9.

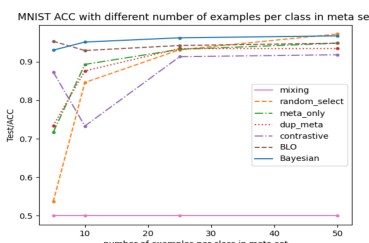

Figure 3: The MNIST test accuracy when trained with meta sets in varying sizes (x-aixs).

In this scenario, the loss function is defined using the standard binary cross-entropy loss. Following [41], all classification models are LeNet5 [32]. The training is conducted on a single GPU, using SGD with a fixed learning rate of 1e-3 and a mini-batch size of 100, over a total of 15,000 steps. In *BADS*, the weight network is implemented as a single-layer Feedforward Neural Network (FNN) with a sigmoid activation function. It takes the top-layer image embeddings from LeNet5 as input

---

[7]`https://github.com/danieltan07/learning-to-reweight-examples.git`

[8]We acknowledge the existence of recent work on DPS with [57, 7] or without [59, 40] meta-data alignment. However, all these studies use two-stage pipelines, where data points are selected offline and then used in the final model training. Since our approach employs an online selection method, i.e. dynamically selecting data while the model is under training, we have only chosen baselines that follow the same style.

and outputs a weight $w_i \in [0, 1]$ for each image. The learning rate for the weight network is 1e-3 and the target sparsity level $\beta$ is 0.005. Other hyperparameters, including those in the baselines, are detailed in Table 3 (Appendix E).

### 3.2.1 Experiment Results and Ablation Study

The classification accuracy is presented in the top-left plot in Figure 2. All approaches, except for *Mixing* and *ClassAct*, achieve over 90% accuracy even when trained with a highly imbalanced *train* set. *Meta_only* demonstrates that training with a small amount of balanced data yields significantly better performance compared to training with a larger but imbalanced training set (*Mixing*). Both *BLO* and *BADS* outperform non-DPS baselines in terms of both accuracy and convergence speed, with *BADS* further outperforming *BLO* by a noticeable margin. *CDS* underperforms all the non-DPS baselines, which we believe is due to the mini-batch-level discrete data selection. When dealing with an extremely imbalanced *train* set, it is possible that the minority class might not be present in some mini-batches. Under these circumstances, the model is compelled to learn from the top examples from the majority class in each mini-batch, potentially leading to biased training outcomes. The top row in Figure 13 and the left plot in Figure 14 (Appendix F) visualizes the weights assigned to the examples in each mini-batch by DPS approaches. All methods, except for*ClassAct*, effectively assigns higher weights to the minority class than to the majority class, thereby directing the classifiers to focus more on the minority class in training.

We further evaluate the models' performance using *meta* sets $\mathcal{D}_m$ of various sizes, with 5, 10, 25, and 50 examples per class included.[9] As illustrated in Figure 3, with a very limited number of meta examples (5 per class), only *BADS* and *BLO* achieve over 90% accuracy on the balanced test set. As the number of available meta data increases, *BADS* consistently leads in performance. However, when sufficient meta data is provided, the gap between *BADS* and the other approaches narrows.

### 3.3 Scenario 2: Data Denoising (CIFAR)

In this scenario, we evaluate the model's ability to manage a noisy *train* set $\mathcal{D}_t$. Although the models are trained using a dataset with significant noise, they are anticipated to perform well on a clean test set. Our experiment utilizes the standard CIFAR 10-class classification dataset [30]. Following the standard CIFAR train/validation set split ratio, we first create a clean and balanced *meta* set $\mathcal{D}_m$ by randomly sampling 1000 examples from each class in the training data. Then, we use the remaining 40,000 examples to create a noisy *train* set $\mathcal{D}_t$ by introducing noise based on the symmetric noise injection setup described in [37, 8, 21]. We set the noise ratio to 0.5 and use the original CIFAR-10 test set in testing.

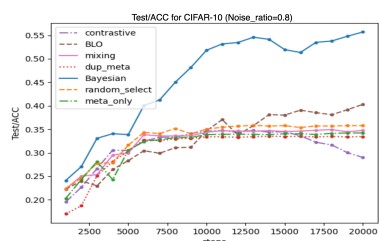

Figure 4: The CIFAR test accuracy when trained with 80% noisy data.

We use ResNet32 [23] as the backbone classification model. Training is performed on a single GPU using SGD with a fixed learning rate of 1e-1 and a mini-batch size of 120 over 20,000 steps. The loss function is the standard multi-class cross-entropy loss. For BADS, the weight network structure retains the same as in scenario 1. However, in this scenario, data point weighting takes into account both the image and its associated label. We represent each label using a one-hot embedding, which is then concatenated with the top-layer image embeddings from ResNet32 and fed into the weight network. The learning rate for the weight network is set to 1e-4. Note that in Eq 7, the gradient of $\theta$ become large if $N_t$ is big. Therefore, we reduce the learning rate of the backbone classification model to 1e-4. The target sparsity level $\beta$ is set to 0.8.[10]

### 3.3.1 Experiment Results and Ablation Study

The classification accuracy is shown in the top-middle plot in Figure 2. All methods, except for the *BLO* approach[11], manage to achieve over 60% accuracy, even using a *train* set contaminated by 50%.

---

[9]Since *ClassAct* uses a similar approach to *CDS* but performs worse, we exclude it from this ablation study.

[10]Besides the primary results, we also present results for asymmetric noise in the top right plot of Figure 11 (see Appendix F.1). Additionally, we explore performance variations by substituting the weight network with the individual weights strategy outlined in [41] (refer to Appendix F.2).

[11]The convergence of *BLO* is slower than that of other approaches. To further explore this, we conduct an ablation study to compare the learning curves for all methods in Appendix F.1.

Notably, *CDS*, *ClassAct* and *BADS* deliver the highest three performances, with *BADS* surpasses all other methods by a noticeable margin. The bottom row of Figure 13 and the middle plot of Figure 14 (Appendix F) visualizes the weights allocated to the examples in each mini-batch by DPS approaches.

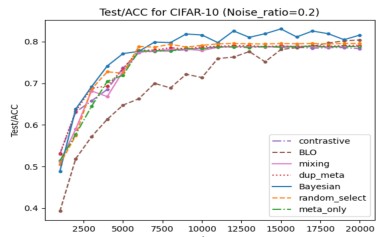

All methods, except for *ClassAct*, consistently gives lower weights to the noisy examples, guiding the classifiers to disregard the noisy examples during training. *ClassAct* assigned lower weights to the noisy examples during the initial stages of training. However, these weights exceeded those of the clean examples in the later phases of training. Interestingly, this behavior did not significantly impact the model's performance.

If we raise the noise label ratio in the *train* set to 80% (Figure 4), both *BLO* and *BADS* still lead the performance, with *BADS* exceeds *BLO* and non-DPS approaches by 15% and 20% in classification accuracy, respectively. *CDS* does not exceed the

Figure 5: The CIFAR test accuracy when trained with 20% noisy data.

performance of non-DPS approaches and starts to overfit on the noisy data after 15,000 training steps. When we lower the noise label ratio to 20% (Figure 5), all methods achieve a classification accuracy of around 80%. Although *BADS* continues to outperform both DPS and non-DPS approaches, the lead is narrower.[12]

### 3.4 Scenario 3: Efficient Learning (WebNLG)

In this scenario, we assess the model's ability to adapt to new domains with very few data points. To demonstrate the robustness of BADS across various research topics and backbone models, we focus on the Natural Language Generation (NLG) task in this section. NLG [18, 12] aims to accurately generate textual descriptions from input tuples. An example is shown in Figure 7 (Appendix D). The English benchmark introduced in WebNLG 2020 [6] includes a training set spanning 16 domains and a test set covering 3 different domains (for details, check Appendix D). We use the original training set (14,239 examples) as our *train* set $\mathcal{D}_t$, and create a single clean and balanced *meta* set $\mathcal{D}_m$ by randomly sampling 30 examples from the WebNLG 2020 validation set in each test domain.

All backbone models are the encoder-decoder T5-small [26]. Training is on a single GPU using Adam with a fixed learning rate of 3e-5 and a mini-batch size of 20 over 40,000 steps. The loss function is the standard negative log likelihood loss. In BADS, the weight network structure remains the same as in scenario 1. Its input is the embedding of the input sequence, represented by the contextual embedding of the "[EOS]" token, and the learning rate is set to 1e-4. The sparsity level $\beta$ is 0.05.

### 3.4.1 Experiment Result

The BLEU scores are displayed in the top-right plot of Figure 2. *BADS* leads the performance, achieving a 2 BLEU score advantage over the second-best system, *Duplicate_Meta*, and surpassing the remaining systems by more than 5 BLEU scores. The other three DPS approaches do not distinguish themselves from the non-DPS methods.

In this scenario, we opt for a more controlled examination of data selection effectiveness due to the big amount of domains in the *train* set[13]. We select three occupation-centric domains—*Athlete*, *Politician*, and *Astronaut*—as our test domains. Additionally, we create a *meta* set $\mathcal{D}_m$ by randomly selecting 50 examples in each of these domains

| Methods | Mem (MB) | Time (s) |
|---|---|---|
| Base | 9184.4 | 61.2 |
| BLO | 22361.3 | 113.48 |
| CDS | 9183.9 | 62.0 |
| -weight calc | – | +700.0 |
| ClassAct | 34821.36 | 269.81 |
| AskLLM-O | 32908.89 | 115.24 |
| -LLM call | – | +13932 |
| BADS | 14694.58 | 61.03 |

Table 1: The average GPU memory and time usage over 100 steps. "Base" represent all non-DPSs.

from their WebNLG 2020 validation set. We then choose two distinct domains, *Artist* and *City*, for training. *Artist*, as another occupation-centric domain, shares a similar schema and vocabulary with the test domains, whereas *City* does not. Ideally, DPS should prioritize the *Artist* examples over those from *City*. The BLEU scores for text descriptions generated by the DPS methods are shown in Figure 12 (Appendix F). These results reinforce the findings from the original experimental setup, with *BADS* outperforming both *BLO* and *CDS* by over 10 BLEU scores. The weights given to the

---

[12]*ClassAct* uses a similar approach to *CDS* and has similar results. We exclude it from this ablation study.

[13]Given that the *train* set spans 16 domains, it is difficult to assess the adequacy of the data selection behavior. We still show the weighting plots for the 16 domain in the bottom row of Figure 13 in Appendix F.

examples in each mini-batch are visualized in the bottom row of Figure 2. *BADS* effectively prioritizes the examples in the *Artist* domain. Instead, *BLO* fails to differentiate between the two domains, and *CDS* incorrectly weights the opposite domain higher. This illustrates that the effectiveness of DPS is linked to the overall performance of the models.

### 3.4.2   Latency

We evaluate the average GPU memory and time consumption using the WebNLG task. Table 1 shows that the training time for *BADS* and *CDS* is similar to that of non-DPSs, while *BLO* and *AskLLM-O* takes nearly twice as long, and *ClassAct* takes even longer. Note that *CDS* and *AskLLM-O* requires additional time to weight examples in the *train* set. Given that the WebNLG 2020 *train* set consists of 35,426 examples, we execute the weighting with a batch size of 20, *CDS* takes a total of 700 seconds. For larger-scale tasks, such as foundation model fine-tuning (Section 4), the weighting process is estimated to consume approximately 22 hours per 1 million training examples. The offline scoring in *AskLLM-O* takes around four hours in our setup on a single NVIDIA A40 GPU. In terms of memory usage, *CDS* aligns with non-DPS approaches, while *BADS* and *BLO* require approximately 1.5 and 2.5 times more GPU memory, respectively. *ClassAct* and *AskLLM-O* takes even more GPU memory.

## 4   Use Case: Large Language Model Instruction Fine-tuning

Instruction Fine-tuning (IFT) for LLMs is a practical application where all three mentioned scenarios are encountered simultaneously. IFT data can be acquired through prompting LLMs [52, 48, 56], gathering existing Natural Language Processing (NLP) benchmarks [35, 54, 50, 43, 53, 38, 56], or employing human annotation [52, 63, 48]. Noise is likely to accumulate during each of these data collection methods. Furthermore, since NLP benchmarks often vary greatly in size, the IFT data typically lacks

| Methods | MMLU | ARCc | ARCe | HellaSwag |
|---|---|---|---|---|
| Mixing | 25.16 | 33.79 | 64.65 | 51.97 |
| Meta_Only | 25.51 | 32.08 | 52.23 | 52.07 |
| Random_Select | 25.62 | 28.92 | 66.75 | 51.83 |
| Duplicate_Meta | 25.30 | 33.28 | 65.15 | 52.76 |
| CDS | 25.62 | 21.16 | 60.14 | 51.68 |
| ClassAct | 24.90 | 31.91 | 67.34 | 52.09 |
| AskLLM-O | 25.55 | **35.15** | 66.88 | **53.71** |
| BADS | **26.59** | 34.39 | **67.00** | 52.91 |

Table 2: Test accuracy of LLMs across four popular benchmarks in eval-harness [17]. Checkpoint selection is using next token prediction accuracy as the selection metric. *Mixing* represents standard IFT.

balance. Additionally, the IFT data does not include data points from the downstream tasks, leading to domain shift in testing.

We use the same IFT data as [57, 51] as our *train* set $\mathcal{D}_t$, which is a mix of FLAN V2 [35], COT [54], DOLLY [10], and OPEN ASSISTANT 1 [31]. Following [57, 5], we focus on four downstream tasks: MMLU [24], which consists of multiple-choice questions across 57 sub-tasks, ARC-challenge/-easy [9], and HellaSwag [61]. Following [57], 5 examples were selected from each sub-task to create the *meta* set $\mathcal{D}_m$ for MMLU, totaling 285 examples. Additionally, following [5], for the other tasks, we randomly chose 25 examples from their validation set to create the respective *meta* sets. To facilitate checkpoint selection, we additionally create a validation set of equivalent size to the *meta* sets.

Due to limited computational resources, we use OpenLLaMA 3B [14] as the backbone model. Training uses one A40 GPU utilizing Adam with a fixed learning rate of 3e-5 and a mini-batch size of 3. In BADS, while the weight network remains the same as described in scenario 3, we modify the input to be the average contextual embedding of all tokens in the sequence. The sparsity level $\beta$ is 0.05. Results are presented in Table 2. We excluded *BLO* from this experiment due to their prohibitive GPU memory usage. Given that the training data for the *Mixing* baseline predominantly consists of IFT data points, it is considered as a standard IFT process. Table 2 indicates that while individual methods may excel in specific tasks, none of the non-DPS baselines nor the *CDS* and *ClassAct* consistently surpass the others across all downstream tasks. However, *BADS* stands out by consistently outperforming all other baselines across every task, except for AskLLM-O, which, as indicated in Table 1, demands significantly more computational resources. According to Table 4 (Appendix F) BADS prefers the human-written-from-scratch, OPEN ASSISTANT 1 and DOLLY, over the rest two created using existing benchmarks.

---

[14]https://huggingface.co/openlm-research/open_llama_3b_v2

To facilitate the use of our proposed datapoint selection method, we provide comprehensive guidelines for hyperparameter tuning in Appendix E. We also include an in-depth discussion and ablation study on the influence of hyperparameters on the method's effectiveness.

## 5   Related Work

Recent works on DPS broadly fall into four categories: i) Approaches based on meta learning (or BLO), ii) Gradient-based methods, iii) Methods based on domain adaptation and transfer learning methods, and iv) General sample reweighing strategies.

• **Meta learning (BLO) approaches.** The DPS can be formulated as a meta learning BLO problem where the outer loss is defined with the training data selection variables, and the inner optimisation is to minimize the model's loss with the selected data [19, 15, 41, 45, 62]. These methods vary in the choice of their outer optimization variables: either directly using the data point weights [41, 45, 19] or mini-batch samplers [15]. To solve the BLO, most approaches rely on computing meta-gradients via reverse mode differentiation [45, 62] while some works utilised reinforcement learning techniques [15]. All these BLO-based methods are computationally demanding with large memory footprint, hindering them from being applied to large-scale models/data.

• **Gradient-based methods.** The key idea is to measure the importance of training data points based on the alignment scores between the loss gradients on training and meta data points. The rationale behind the gradient alignment can be theoretically underpinned by the BLO perspective. As shown in [41], the one-step inner loss update with zero initial weights in BLO reduces to the gradient alignment (cosine angle) between the train and meta data points [57]. However, the method in [57] requires evaluating and storing gradients of the entire training data points, thus computationally expensive. Furthermore, the final solution may not be optimal since the gradient computation is done with an initial network that is warm-up trained with a random subset of the training data. In [16] the weights are optimised by the expected gradient norms during the network training, which serves as a proxy for the importance of data points. In [28] they find the subsets that closely match the gradient of the training and meta sets using an orthogonal matching pursuit algorithm. Under online continual learning setup [2], they formulate sample selection as a constraint reduction problem based on the constrained optimization view of continual learning.

• **Domain adaptation and transfer learning methods.** Selecting a subset of the train set that are best aligned with the in-domain meta set can be naturally seen as *domain adaptation* and *transfer learning*. In the NLP community, [1] finds that the large language models implicitly learn sentence representations that cluster by domains without supervision, while in [20], they show the effectiveness of domain-adaptive pre-training with data augmented using simple data selection strategies. In [39] they proposed domain adaptive transfer learning which computes the importance weights on the target dataset from the ideas in domain adaptation and estimation of the label distribution. In multi-task transfer learning community, some previous works identify the detrimental effects of the gradients from different tasks, and propose strategies to mitigate the conflicting gradients issue [60, 11, 34].

• **General sample reweighing strategies.** Since DPS essentially involves finding the optimal reweighed distribution with the underlying domain itself unchanged, several approaches aim to tackle the problem via importance sampling techniques. In [27] they derive a tractable upper bound to the per-sample gradient norm, and derive an estimator of the variance reduction achieved with importance sampling. The curriculum learning [3] is also closely related as one can design an optimal schedule of the successive training distributions. For instance, in [44] they introduce the so-called data parameters, associating samples and classes in the train data with learnable parameters, which governs their importance in the learning process within the curriculum learning framework.

## 6   Conclusion

We revisited the DPS problem from the perspective of Bayesian learning. By treating the neural network of interest, as well as an auxiliary weighting neural network as random variables and inferring their joint posterior using SGLD, we achieve a simple and effective approach to data reweighting that is more reliable and scalable than BLO alternatives. Our framework is straightforwardly applicable to learning with data imbalance, label noise, and automating auxiliary data curation. We demonstrate our framework can apply to automating curation of the wide variety of auxiliary instruction fine-tuning data available for billion-scale language models. Overall this demonstrates a promising new kind of approach to the growing need for data optimization in neural network learning.

# Limitations

Our proposed *BADS* algorithm has the following limitations, which we plan to address as our future work.

1. There are several hyperparameters involved, which need to be carefully tuned for best performance. These include: the sparsity level parameter $\beta$, the relative impact constants $\rho$'s of the objective terms in our SGLD update (detailed in App. E), and the standard hyperparameters (e.g., batch size, learning rates, weight decay). Some of these hyperparameters may be optimised via Bayesian model selection in a principled manner. For instance, the sparsity level $\beta$ can be regarded as a latent random variable (a part of the model) with a proper prior distribution imposed on it, and we can do posterior inference of $\beta$ (or marginalise it out) together with $w$ and $\theta$ in our SGLD update equations. We will pursue this in our future study.

2. Although our method is computationally far more efficient than BLO and some other DPS approaches, its GPU memory footprint is demanding compared to non-DPS algorithms. This mainly originates from the entire weight variables or the whole weight network parameters loaded/maintained in the memory for frequent updates. One possible workaround is to load only the weight variables that are assoicated with the current minibatches. We will be investigating this code optimisation further in our ongoing future study.

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

# Appendix

## A Posterior Derivation for Eq. 4

From the graphical model in Fig. 1, we exploit the conditional independence $D_m \perp (w, D_t) \mid \theta$.

$$p(\theta, w | D_t, D_m) = \frac{p(\theta, w, D_m | D_t)}{p(D_m | D_t)} \tag{10}$$

$$= \frac{1}{p(D_m | D_t)} \cdot p(w) \cdot p(\theta, D_m | w, D_t) \tag{11}$$

$$= \frac{1}{p(D_m | D_t)} \cdot p(w) \cdot p(D_m | \theta, w, D_t) \cdot p(\theta | w, D_t) \tag{12}$$

$$= \frac{1}{p(D_m | D_t)} \cdot p(w) \cdot p(D_m | \theta) \cdot p(\theta | w, D_t) \tag{13}$$

$$\propto \cdot p(w) \cdot p(D_m | \theta) \cdot p(\theta | w, D_t) \tag{14}$$

In (13) we use $D_m \perp (w, D_t) \mid \theta$, and in (14) $\frac{1}{p(D_m | D_t)}$ is regarded as constant for it has nothing to do with $\theta$ and $w$.

## B Posterior Derivation for Weight Network Cases

Here we provide posterior derivation for the weight network case $w_i = w(z_i^t; \phi)$. First, the weights become a deterministic function of $D_t$ and $\phi$ (here $\phi$ = weight network parameters) as shown in Fig. 6(a). But since $w$ is a deterministic function of $\phi$ and $D_t$, we can simplify it by having $w$ absorbed into $\phi$ while introducing conditional dependence (an arrow) from $D_t$ to $\phi$, as depicted in Fig. 6(b). Note that $D_t$ is always given, we treat $\phi$ as a random variable, and wherever $w_i$ appears, we replace it by $w(z_i^t; \phi)$. More specifically, we make the following changes to the equations in the weight net scenario:

Eq. 2: $p(\theta | \phi, D_t) \propto p(\theta) \cdot \prod_{i=1}^{N_t} p(w(z_i^t; \phi), z_t^i | \theta)$

Eq. 8: $p(\phi | D_t) \propto e^{-(\sum_i w(z_i^t; \phi) - \lfloor N_t \beta \rfloor)^2 / 2\sigma^2}$

Eq. 4 (with detailed derivations):

$$p(\theta, \phi | D_t, D_m) = \frac{p(\theta, \phi, D_m | D_t)}{p(D_m | D_t)} \tag{15}$$

$$= \frac{1}{p(D_m | D_t)} \cdot p(\phi | D_t) \cdot p(\theta, D_m | \phi, D_t) \tag{16}$$

$$= \frac{1}{p(D_m | D_t)} \cdot p(\phi | D_t) \cdot p(D_m | \theta, \phi, D_t) \cdot p(\theta | \phi, D_t) \tag{17}$$

$$= \frac{1}{p(D_m | D_t)} \cdot p(\phi | D_t) \cdot p(D_m | \theta) \cdot p(\theta | \phi, D_t) \tag{18}$$

$$\propto \cdot p(\phi | D_t) \cdot p(D_m | \theta) \cdot p(\theta | \phi, D_t) \tag{19}$$

Eq. (5): $[\theta, \phi] \leftarrow [\theta, \phi] + \frac{\eta}{2} \nabla_{\theta, \phi} \log p(\theta, \phi | \mathcal{D}_t, \mathcal{D}_m) + \epsilon \sqrt{\eta}, \quad \epsilon \sim \mathcal{N}(0, I)$

Eq. (6): $\theta \leftarrow \theta + \frac{\eta}{2} \nabla_\theta \left( \log p(\theta) - N_t \cdot \mathbb{E}_{i \sim \mathcal{B}_t} \left[ w(z_i^t; \phi) \cdot l(z_i^t; \theta) \right] - N_m \cdot \mathbb{E}_{j \sim \mathcal{B}_m} \left[ l(z_j^m; \theta) \right] \right) + \epsilon_\theta \sqrt{\eta}$

Eq. (7): $\phi \leftarrow \phi + \frac{\eta}{2} \nabla_\phi \left( \log p(\phi | D_t) - N_t \cdot \mathbb{E}_{i \sim \mathcal{B}_t} \left[ w(z_i^t; \phi) \cdot l(z_i^t; \theta) \right] \right) + \epsilon_\phi \sqrt{\eta}$

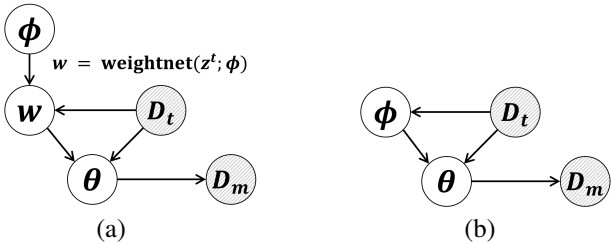

(a)             (b)

Figure 6: Graphical models when the weight network is adopted. (a) One possible representation. (b) More simplified representation by absorbing $w$ into $\phi$ using deterministic $w = \text{weightnet}(z_i^t; \phi)$. See texts for details.

## C Convergence Analysis

**Assumption C.1** (Adjusted from Assumption 4.3 in [64]). *There exists $m > 0$ and $b \geq 0$ such that*

$$\left\langle \nabla_{\theta, w} \log p(\theta, w | D_t, D_m), \begin{bmatrix} \theta \\ w \end{bmatrix} \right\rangle \geq m \left\| \begin{bmatrix} \theta \\ w \end{bmatrix} \right\|_2^2 - b \tag{20}$$

*holds for any $\theta, w$.*

**Assumption C.2** (Adjusted from Assumption 4.4 in [64]). *Any minibatch gradient of the log-posterior is Lipschitz continuous. That is, there exists a constant $L$ such that for any $z_i^t \in D_t$ and $z_j^m \in D_m$,*

$$\left\| \nabla_{\theta, w} \Big( \log p(w) + \log p(\theta) - N_t w_i l(z_i^t; \theta) - N_m l(z_i^m; \theta) \Big) - \right.$$
$$\left. \nabla_{\theta', w'} \Big( \log p(w') + \log p(\theta') - N_t w_i l(z_i^t; \theta') - N_m l(z_i^m; \theta') \Big) \right\|_2 \leq L \left\| \begin{bmatrix} \theta \\ w \end{bmatrix} - \begin{bmatrix} \theta' \\ w' \end{bmatrix} \right\|_2 \tag{21}$$

*holds for any $\theta, w, \theta', w'$.*

**Theorem C.3** (Adjusted from Theorem 4.5 in [64]). *Let $d = \dim(\theta) + N_t$, $B$ be the batch size, and $\rho$ be the Cheeger constant. For any $\epsilon \in (0, 1)$, with the initial iterate satisfying $p\left( \left\| \begin{bmatrix} \theta^{init} \\ w^{init} \end{bmatrix} \right\| \leq R/2 \right) \leq \epsilon/16$ for $R = \overline{R}(\epsilon K^{-1}/12)$, and step size $\eta = \tilde{O}(\min\{\rho^2 d^{-2}, B^2 \rho^2 d^{-4}\})$, the distribution $\mu_K^{SGLD}$ of the $K$-th iterate in our SGLD iterations Eq. (6–7) satisfies:*

$$\left\| \mu_K^{SGLD} - p(\theta, w | D_t, D_m) \right\|_{TV} \leq \lambda (1 - C_0 \eta)^K + B^{-1} C_1 \eta^{1/2} + C_2 \eta^{1/2} + \epsilon/2 \tag{22}$$

*for some constant $\lambda > 0$, $C_0 = \tilde{O}(\rho^2)$, $C_1 = \tilde{O}(Rd\rho^{-1})$, $C_2 = \tilde{O}(d\rho^{-1})$. Here $\| \cdot \|_{TV}$ stands for the total variation distance, and $\overline{R}$ is defined as:*

$$\overline{R}(z) = \max \left\{ \frac{625 d \log(4/z)}{m}, \frac{4d \log(4L/m) + 4b}{m}, \frac{4d + 8\sqrt{d \log(1/z)} + 8 \log(1/z)}{m} \right\}^{1/2}. \tag{23}$$

## D Details of the Tasks

**WebNLG 2020**    We use the release_v3.0_en version of WebNLG benchmark. Domains in training set are: *Building, Astronaut, City, University, MeanOfTransportation, SportsTeam, Food, Artist, Company, ComicsCharacter, Monument, Airport, Politician, Athlete, WrittenWork, CelestialBody.* Domains in test set are *MusicalWork, Scientist, Film.*

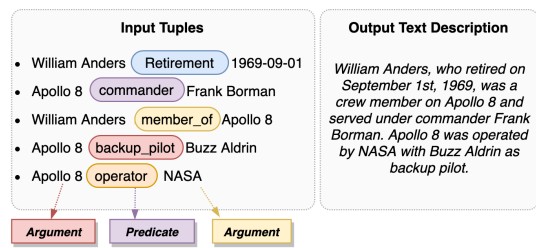

Figure 7: An example of Natural Language Generation.

# E  Details of the Experiments

## E.1  Hyperparameters Details

To enhance flexibility in managing the training process, we incorporate $\rho_\theta^t$, $\rho_\theta^m$, and $\rho_w^t$ into Eq 7 as follows:

$$\theta \;\leftarrow\; \theta + \frac{\eta}{2}\nabla_\theta\Big(\log p(\theta) - \rho_\theta^t N_t\cdot\mathbb{E}_{i\sim\mathcal{B}_t}\big[w_i\cdot l(z_i^t;\theta)\big] - \rho_\theta^m N_m\cdot\mathbb{E}_{j\sim\mathcal{B}_m}\big[l(z_j^m;\theta)\big]\Big) + \epsilon_\theta\sqrt{\eta} \tag{24}$$

$$w \;\leftarrow\; w + \frac{\eta}{2}\nabla_w\Big(\log p(w) - \rho_w^t N_t\cdot\mathbb{E}_{i\sim\mathcal{B}_t}\big[w_i\cdot l(z_i^t;\theta)\big]\Big) + \epsilon_w\sqrt{\eta} \tag{25}$$

The training hyperparameters for BADS and CDS are shown in the table below. Other hyperparameters have been shown in the main paper.

| BADS | $\eta$ | $\rho_\theta^t$ | $\rho_\theta^m$ | $\rho_w^t$ | $\sigma$ | $\beta$ | $s_{avg}$ | CDS | $r_{floor}$ | H | lr_halv |
|---|---|---|---|---|---|---|---|---|---|---|---|
| MNIST | 1.0 | 1.0 | 1.0 | 1.0 | 5e-5 * $N_t$ | 0.005 | 10 | MNIST | 0.005 | 1000 | 5000 |
| CIFAR | 1.0 | 0.1 | 1.0 | 1.0 | 5e-5 * $N_t$ | 0.8 | 10 | CIFAR | 0.8 | 15000 | 10000 |
| WebNLG | 1.0 | 1.0 | 1.0 | 1.0 | 1e-5 * $N_t$ | 0.05 | 10 | WebNLG | 0.05 | 6700 | 20000 |
| LLMs | 1.0 | 0.5 | 1.0 | 1.0 | 1e-3 * $N_t$ | 0.05 | 10 | LLMs | 0.05 | 4200 | 15000 |

Table 3: Hyperparameters for all experiments.

## E.2  Guildline for Hyperparameters Tuning

There are two primary sets of hyperparameters in *BADS*:

- hyperparameters for model's parameters update: $\eta$, $\epsilon_\theta$, $\epsilon_w$, $\rho_\theta^t$, $\rho_\theta^m$, and $\rho_w^t$ (Eq. 24-25),
- hyperparameters for the prior distribution of the example weights, which are $\sigma$, $\beta$, $s_{avg}$ (Eq. 8-9).

In general,

- we set $\eta$ to 1 and kept the Gaussian noise small with $\epsilon_\theta$ and $\epsilon_w$ equal to 1e-5;
- $\rho_w^t$ is set to 1;
- we set $\rho_\theta^m$ to 1 and primarily adjust $\rho_\theta^t$;
- in most cases, $\rho_\theta^t$ is simply set to 1. However, if the training set contains noise—where the ground truth labels might be incorrect—the loss from the training examples, particularly in the early stages of training, may be unreliable. In such cases, we decrease $\rho_\theta^t$;
- $s_{avg}$ should not be too large, as it may incorporate outdated weights from earlier training steps. We set it to 10;
- we select $\beta$ based on the proportion of training data we consider beneficial for the downstream tasks. For the LLMs Instruction Fine-tuning experiments, we adopt the same ratio used in the previous studies [52, 46].
- $\sigma$ controls how tightly the weights should match $\beta$. We set $\sigma$ based on our confidence in the selection ratio $\beta$: a smaller $\sigma$ indicates greater confidence in $\beta$.

- data denoising scenario is a bit special, Eq 7 shows that high losses from the training batch push the example weights $w$ toward 0. With a high noise rate (50% and 80%), the training batch losses remain high throughout the training process. To prevent the weights $w$ from collapsing to 0, we use a high selection ratio $\beta$ and a low $\sigma$.

### E.3 Ablation study: Influence of the impact constants $\sigma$ and sparsity level $\beta$.

**Impact constant $\sigma$**   Higher $\sigma$ causes the example weights $w$ to drift away from $\beta$, occasionally collapsing to 0, and may result in incorrect example weights (see Figure 8). The right column in Figure 11 shows that in all three proof-of-concept scenarios, the models achieve similarly good performance when $\sigma$ is reduced to around $10^{-5}$.

**Sparsity level $\beta$**   In both WebNLG and MNIST scenarios, high $\beta$ leads to incorrect example weights (see Figure 9). Conversely, due to the reason we mentioned above, in data denoising (CIFAR) scenario, lower $\beta$ leads to incorrect weights. The left column in Figure 11 shows that the backbone models' performance significantly declines in both the WebNLG and MNIST scenarios when the example weights are incorrect. In the CIFAR scenario, the impact is less pronounced.

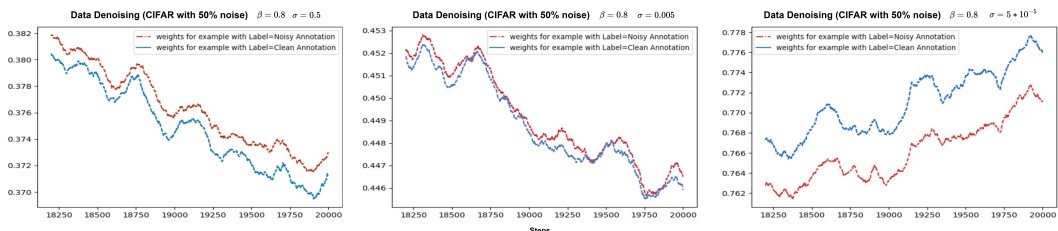

Figure 8: **Varying the impact constant $\sigma$.** The data denoising scenario is conducted on CIFAR-10 with 50% noisy data, while the sparsity level $\beta$ is maintained at 0.8, consistent with the main experiments. From left to right, $\sigma$ decreases from 0.5 to $5 \times 10^{-5}$. When $\sigma = 0.5$, the weighting network assigns higher weights to the noisy examples. At $\sigma = 0.05$, it assigns similar weights to both noisy and clean examples. Only when $\sigma$ is sufficiently low ($\leq 5 \times 10^{-5}$) does the model assign distinctly higher weights to the clean examples.

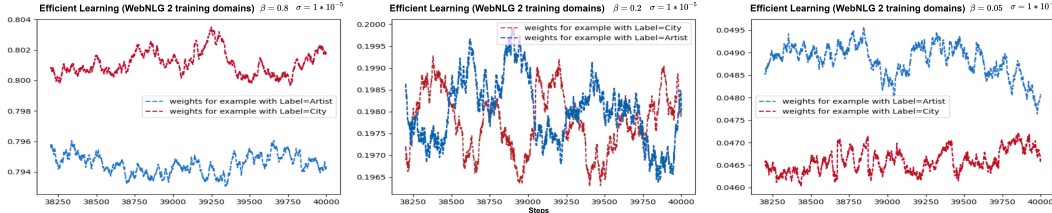

Figure 9: **Varying the sparsity level $\beta$.** This efficient learning scenario is conducted on WebNLG training with two domains: City and Artists. Since Artists has greater overlap in schema and vocabulary with the downstream domains, the model should assign higher weights to examples from this domain. The impact constant $\sigma$ is maintained at $1 \times 10^{-5}$, consistent with the main experiments. As $\beta$ decreases from 0.8 to 0.05, the weighting network behaves as follows: at $\beta = 0.8$, it assigns higher weights to the City domain; at $\beta = 0.2$, it assigns similar weights to both domains; and at $\beta = 0.05$, it assigns significantly higher weights to the Artists domain.

## F   Analysis of Main Results

### F.1   Learning Curves Study

The experimental results discussed in Section 3.3.1 demonstrate that *BLO* converges more slowly compared to other approaches. In this section, we aim to examine the learning curves of DPS approaches, as illustrated in Figure 11. The trends for each approach appear consistent across both

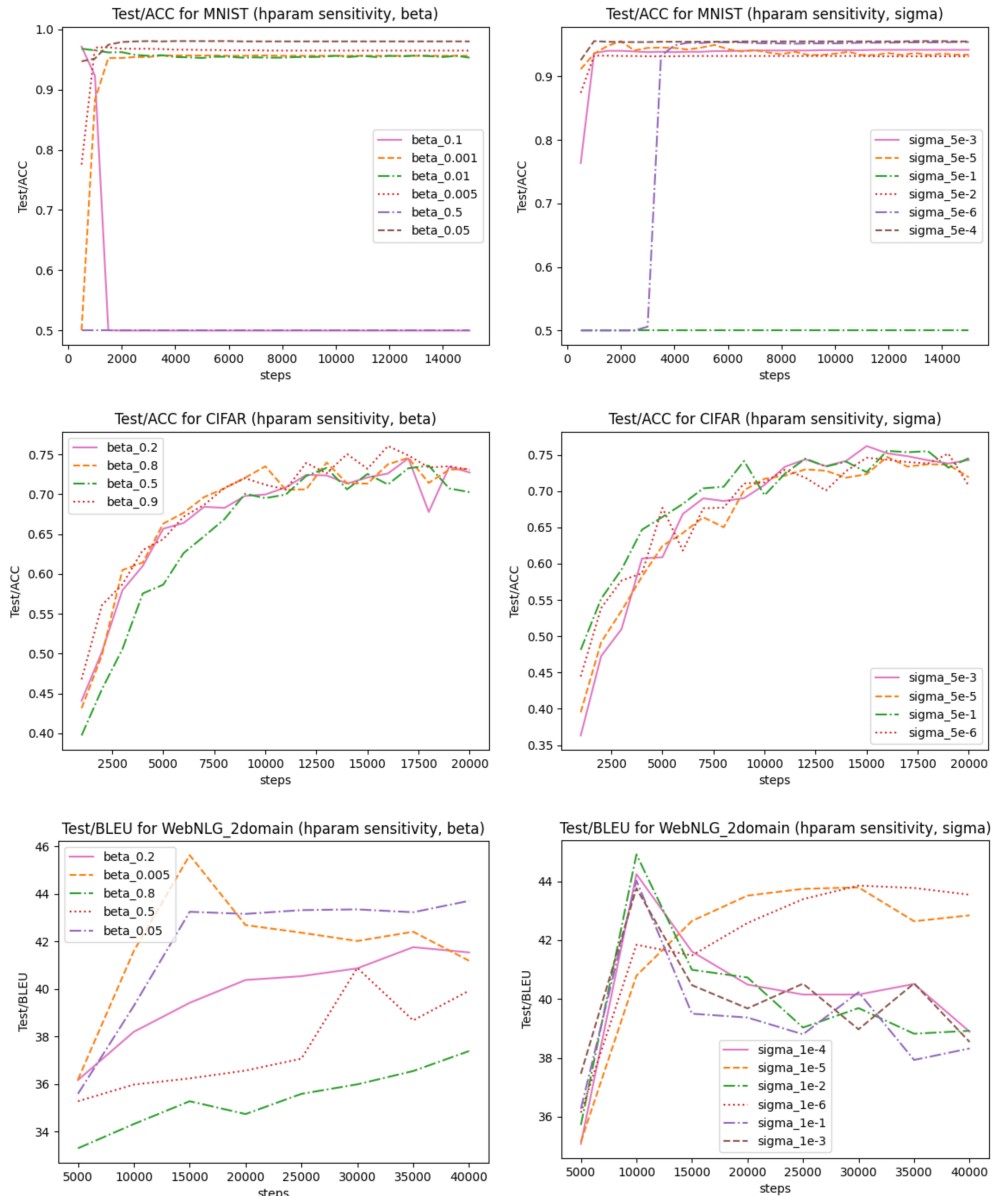

Figure 10: Model's performance in the three proof-of-concept scenarios with different $\beta$ and $\sigma$.

symmetric and asymmetric noise experiments. The convergence rate of *BADS* aligns with that of non-DPS approaches, whereas *CDS* converges faster than *BADS*, and *BLO* converges significantly slower. In the asymmetric noise experiments, overfitting is less pronounced compared to the symmetric noise experiments, where after 50,000 training steps, the test accuracy for all approaches decreases. *BLO* and *BADS* exhibit a notably slower rate of overfitting compared to other methods. Conversely, *CDS* overfits more quickly than non-DPS approaches, resulting in significantly lower test accuracy even compared to the non-DPS methods.

## F.2   Individual Scalar Weights *vs*. Weight Network

In [41], each training example is associated with a learnable scalar weight. The scalability issues of this method compared to the weight network used in *BADS* are detailed in Section 2.4. In this section, we examine the CIFAR-10 denoising task to assess performance differences that result from replacing the *BADS* weight network with the individual weights strategy described in [41]. From the top row of Figure 11, it is evident that the performance of *BADS* with scalar weight (referred to as

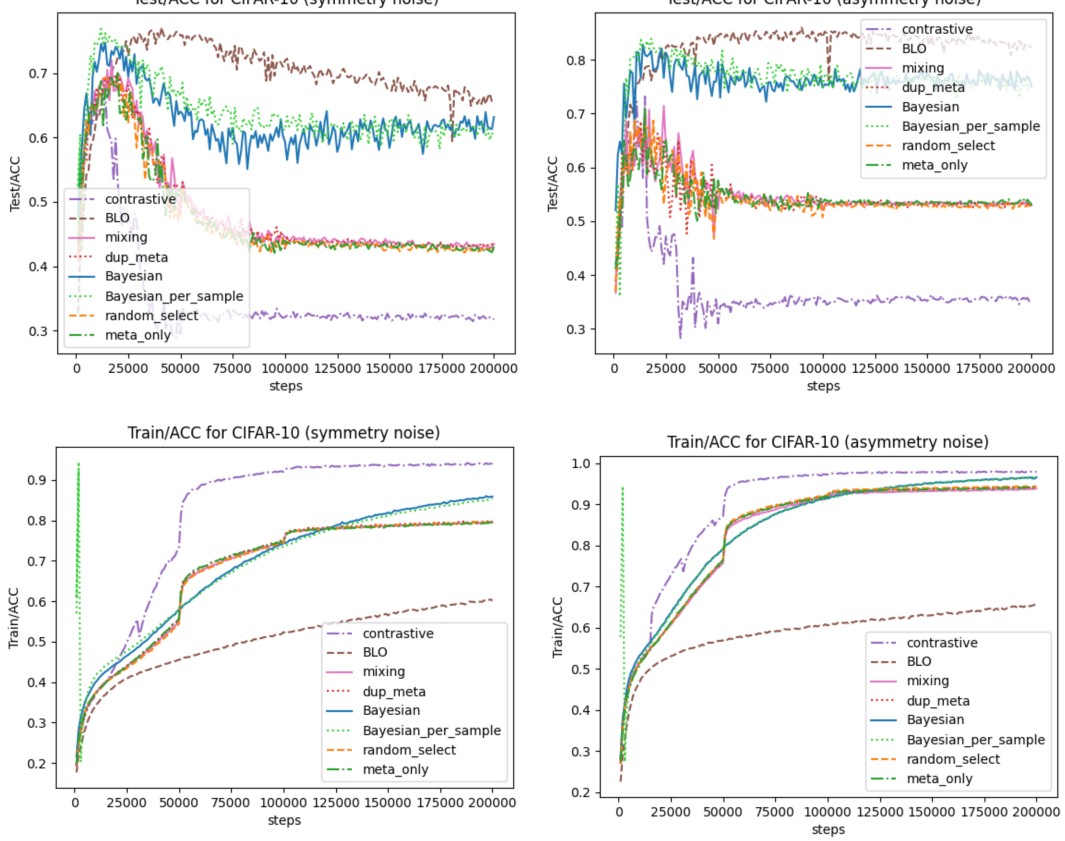

Figure 11: This graph shows the test/train accuracy over 200K training steps. The x-axis denotes the training steps, and the y-axis indicates the accuracy levels. The top row displays the testing accuracy, while the bottom row shows the training accuracy. In the left column all models are trained using *train* set contaminated by 50% symmetric noise. While, in the right column, the *train* set contaminated by 40% asymmetric noise.

*Bayesian_per_sample* in the legend) slightly surpasses that of the original *BADS* (labeled as *Bayesian* in the legend).

| IFT sets | Avg Example Weights |
|---|---|
| OPEN ASSISTANT 1 [31] | 0.0063 |
| DOLLY [10] | 0.0025 |
| Flan-V2 [35] | 0.0023 |
| CoT [54] | 0.0005 |

Table 4: The average scores IFT examples get from BADS.

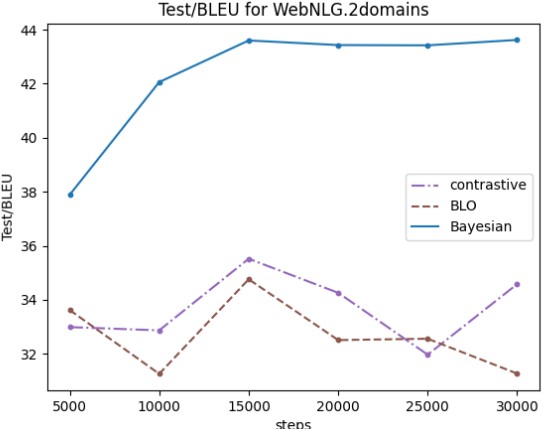

Figure 12: Scenario 3, domain adaptation using WebNLG benchmark. This plot shows the BLEU scores on WebNLG benchmark. All DPS models are trained on 2 domains, *Artist* and *City*, and tested on other 3 domains – *Athlete*, *Politician*, and *Astronaut*. The x-axis represents the training steps, while the y-axis shows the evaluation metric, BLEU.

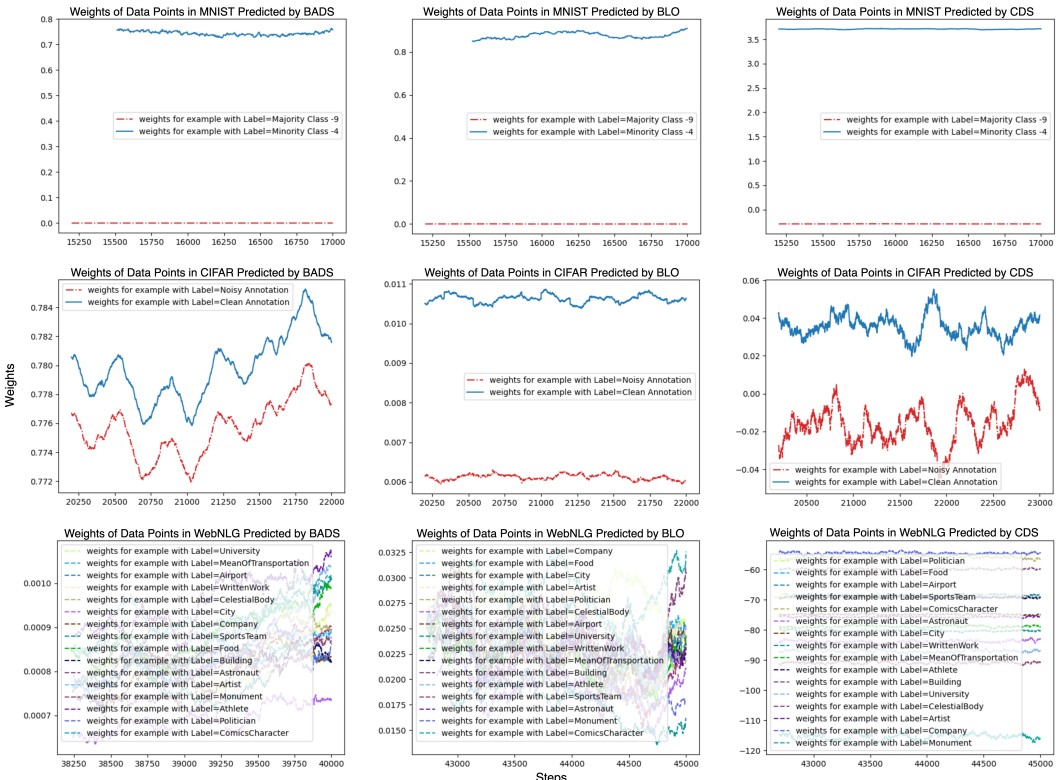

Figure 13: Proof-of-Concept experiment supplementary results. All plots illustrate the average weights of data points within mini-batches during the last 2000 training steps, with the x-axis representing the training steps and the y-axis showing the average weights. Classes depicted in **blue** are expected to receive higher weights compared to those in **red**. The **top row** displays the MNIST experiments from scenario 1, the **middle row** shows the CIFAR experiments from scenario 2, and the **bottom row** features the WebNLG experiments from scenario 3. The **left**, **middle**, and **right columns** correspond to *BADS*, *BLO*, and *CDS*, respectively.

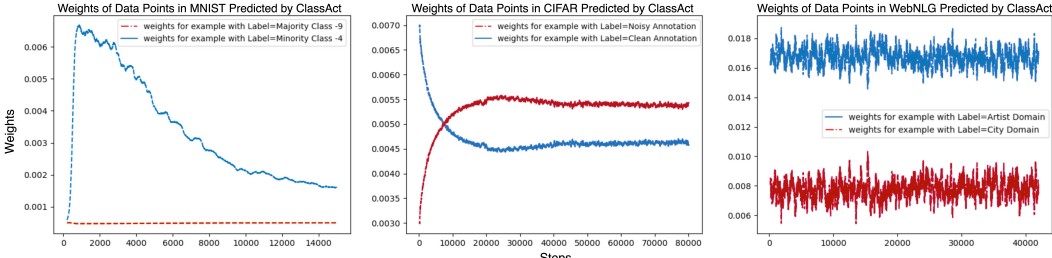

Figure 14: Proof-of-Concept experiment supplementary results. All plots illustrate the average weights of data points within mini-batches during the training of ClassAct method, with the x-axis representing the training steps and the y-axis showing the average weights. Classes depicted in **blue** are expected to receive higher weights compared to those in **red**. The **left** plot displays the MNIST experiments from scenario 1, the **middle** plot shows the CIFAR experiments from scenario 2, and the **right** plot features the WebNLG experiments from scenario 3.

