# OpenReview forum: "A Bayesian Approach to Data Point Selection"
_NeurIPS.cc/2024/Conference — NeurIPS 2024 poster_

### Official Review · Reviewer_5MKv · 2024-07-04

**Soundness:** 2
**Presentation:** 3
**Contribution:** 4
**Rating:** 7
**Confidence:** 3

**Summary:**

The paper proposes a method for data point selection (DPS) in the setting where only little data from the target (or meta) distribution is available, together with a lot of data from the training distribution. DPS aims to find the weights of data points in the training distribution such that the model parameters that minimise the loss of the therewith weighted training distribution also minimise the loss on the target distribution. The method proposed for DPS in this paper consists in treating the problem in a Bayesian way and doing posterior inference on both (data) weights and model parameters simultaneously, using stochastic gradient Langevin dynamic sampling (SGLD) to approximate the posterior. The crucial trick is to view the data from the target distribution as generated by the model (parameters), which allows for the inference.
They show the effectiveness of their approach on 4 experiments and compare against 2 DPS baselines (bilevel optimisation (BLO) and contrastive data selection (CDS)) as well as a range of naive baselines (mixing available training and target distributions to different extents).
- The first experiment is on data balancing for MNIST, where the class balance differs significantly between training (imbalanced) and target (balanced) distribution.
- The second experiment is on data denoising for CIFAR, where the training distribution (noisy) differs from the target distribution (clean) in the degree of noisiness
- The third experiment is on efficient learning on WebNLG, where there are differences in the topic or domain of the data between training distribution (Artist, City) and target distribution (Athlete, Politician, Astronaut)
- The fourth experiment is on instruction finetuning (IFT) for LLMs, using an OpenLlama3B model.
The proposed method compares favourably in all experiments in the main part of the paper. In the appendix results for the second experiment are additionally shown for longer training times, in which case BLO outperforms the proposed method.

**Strengths:**

- The idea of framing and solving DPS in a Bayesian way seems novel and innovative
- The paper is well written in general and is easy to read and follow
- The proposed method is getting good empirical results as displayed in the paper, in particular also on the relevant use case of IFT for LLMs
- Compared to BLO, the proposed method seems to have lower memory and compute requirements

**Weaknesses:**

- For the most part, the method proposed is explained well and in detail. However, when it comes to a crucial aspect in the implementation  that is subsequently used in all experiments (the weight network), details are brushed under the carpet as a 'straightforwardly modification'. However, for me the causal graph in Figure 1 seems to be different if the weights become a function of $z^t$, let alone $\theta$ if model embeddings $f_{\theta}(z^t)$ are used instead of raw data $z^t$ like in the experiments. This should be explained in more detail (see also Question about this below).
- In the main part of the paper, performance of BLO is portrayed as clearly falling behind the proposed method and other methods in terms of accuracy on the CIFAR experiment (Fig 2, Fig 3, line 266). However in Fig 7 in the Appendix, it can be seen that with longer training, BLO is at least on par (symmetry noise) or even slightly better (asymmetry noise) in terms of accuracy. It would be better if this would already be mentioned in the main part of the paper
- Assuming a non-negativity constraint on the weights and the loss, in Eq (4) and subsequently Eq (7) the gradient wrt $w$ will be negative for the data dependent term and thus each gradient update is only pointing away from $0$ through the gradient of the prior term $\log p(w)$ (and the noise term at the end). Therefore, preventing collapse to $0$ weights (as described in line 148) seems to crucially depend on a strong enough prior as well as the impact constant and $N_t$. At least for the not so compute intensive experiments on MNIST and CIFAR, it would be good to see an ablation on the effect of these impact constants and the prior governing sparsity parameter $\beta$.
- The limitations are only spelled out after the page limit

**Questions:**

- Which part in the prior formulation or in the optimisation ensures that the weights are non-negative? Or are they intentionally not constrained by the method in that sense?
- How does the causal graph in Figure 1 change if a weight network that is using embeddings of training data as inputs is being used like in the experiments? How does posterior inference change in this case?
- Why is the weight network being used in the experiments if in the ablation in the appendix, the element wise weights version of the proposed approach is outperforming it?
- Why are the inputs to the weight network different for the MNIST experiment (image embeddings) and the CIFAR experiment (image embedding + one-hot encoded label)?
- When comparing to BLO in terms of compute efficiency, have the authors taken into account proposals of making the Hessian vector product calculations more efficient for BLO (see https://iclr-blogposts.github.io/2024/blog/bench-hvp/ for example)?

**Limitations:**

- Limitations are discussed, but after the page limit

---

> ### Author Rebuttal · Authors · 2024-08-06
>
> ## [Re: Graphical Model and Derivations]
> See the General Rebuttal:
> * [Re-1: Derivations for Eq 4]
> * [Re-2: Graphical Model and Derivations with Weight NN]
>
> ## [Re: Hyperparameter Analysis]
> See [Re-3: Hyperparameter Analysis] in the General Rebuttal
>
> ## [Re: Ensuring the Non-negative Weights]
>
> Lines 205-206 of the paper indicate that in BADS, the weight network is implemented as a single-layer Feedforward Neural Network (FNN) with a sigmoid activation function. The sigmoid function ensures that the output weights are valid. We will highlight this in the revised paper.
>
> ## [Re: Why Weight NN]
>
> In the paper, specifically in lines 170-174, we note that this weight network approach is effective for smoothing and regularizing output weights due to the smooth functional properties of neural networks. Importantly, if additional training samples are added after the model has been trained, the learned weight network can be used to assign weights or select samples from the new dataset without needing to retrain the entire model from scratch.
>
> Additionally, obtaining precise element-wise weights typically requires training the model on the entire training set for multiple epochs, which can be costly, especially for large foundational models fine-tuning. By using the weight network, we do not even need to train the model with one full epoch. After training on a subset of the data for one epoch, the weight network can be employed to assign weights to the remaining examples. We can then select the examples with higher predicted weights from this set and continue training the model on these selected samples. This approach further enhances training efficiency. We plan to conduct related experiments in future work.
>
> We will explain with more details about this in the revised paper.
>
> ## [Re: Different Weight NN for MNIST and CIFAR]
>
> In the MNIST scenario, we focus on data balancing, where the minority and majority examples can be distinguished solely by the input images. In contrast, the CIFAR scenario targets at data denoising. Here, we generated noisy examples by randomly sampling class labels for the input images (L242-245), making it impossible to determine whether an example has a correct label (clean examples) or an incorrect label (noisy examples) just by examining the input image. As a result, the weight network requires both image and label information for accurate decision.
>
> ## [Re: Efficient Hessian Vector Products (HVPs)]
>
> We use PyTorch's (v2.3) automatic differentiation so there is no explicit Hessian computation. Specifically, we use reverse-over-reverse differentiation and hence is possible that we can get more gains using other methods (e.g. reverse-over-forward) for BLO. Nevertheless, regardless of how HVP is computed in practice, BLO is still more expensive than BADS as BADS doesn't require 2nd-order computation.

---

> > ### Comment · Reviewer_5MKv · 2024-08-09
> >
> > I thank the authors for answering some of my questions (non-negativity of weights, different weight NN inputs for MNIST and CIFAR, why using the weight NN).
> >
> > I also acknowledge that they added some ablation on WebNLG about the prior parameters $\beta$ and $\sigma$ as well as a derivation on the inference when a weight network is being used. Regarding this however I have 2 questions:
> > - How does inference change if instead of $z^t$ model embeddings $f_{\theta}(z^t)$ are used like in the experiments of the paper? Does that not cause a circular dependence between $\theta$ and $w$?
> > - Why is the Test/BLEU for $\beta=0.05$ in the ablation at 44 after 40k vs. only 36 in Figure 2 of the paper?

---

> ### Author Response · Authors · 2024-08-09
> **RE: Official Comment by Reviewer 5MKv**
>
> We appreciate the reviewer’s engaging discussion and sharp questions.
>
> **[RE: Circular Dependence]**
>
> Strictly speaking, if we use the embedding $f_\theta(z^t)$ in the weightnet, then weight $w$ becomes a function of both $\theta$ and $D_t$, thus we can have circular dependency as the reviewer pointed out.
>
> However, we would rather see this case in the following way: Suppose we were able to access the ideal oracle embedding function $f^*$. Then ideally the weightnet would use this $f^*$ internally to output the weight $w$, ie, $w = weightnet(f^*(z^t); \phi)$. This way, as $f^*$ is constant, our graphical model in Fig.1(b) of the attached PDF is still valid, and there is no circular dependency. In practice, however, since we cannot access $f^*$, one reasonable way to approximately mimic it is to use the current estimate $f(z^t; \theta)$ as an *online plugin proxy for* $f^*$. We hope this answers your question. FYI, in the implementation, we do stop-gradient $\theta$ in the weightnet update, which is in line with our argument of *online plugin proxy for* $f^*$.
>
> **[RE: Test/BLEU]**
>
> For WebNLG, we did experiments on two different setups:
> * **The main setup** (the performance is shown in *top-right plot in Figure 2 of the paper*): We use the original WebNLG domain adaptation set up (16 domains for training and 3 domains for testing) to report the models generation performance. Details are explained in *Appendix A*.
> * **Ablation setup (called 2-domain)** (the performance is shown in *Figure 8 in the Appendix*): The purpose of this setup is to easily assess the effectiveness of data selection, as visualizing the data selection across 16 domains can be challenging. Details and the rationale for this setup are explained in lines 291-301 of the paper.
>
> Generally, the Ablation setup is somewhat simpler than the main setup, resulting in a higher test BLEU score.
>
> In the Hyperparameter Analysis section of the rebuttal, we did experiments using the Ablation setup to visualize how hyperparameters affect data selection behavior (shown in Figure 3 in the rebuttal PDF). So, reviewers comparing Figure 1 in the rebuttal PDF to Figure 8 in the paper will observe that the test BLEU score for $\beta=0.05$ consistently hovers around 44.
>
> If the reviewer has any further questions, please feel free to reach out.

---

> > ### Comment · Reviewer_5MKv · 2024-08-12
> >
> > Thanks, I understand now the discrepancy between Figure 2 in the paper and the ablation in the rebuttal pdf. I definitely think some discussion around the hyper parameter sensitivity based on the ablations in the rebuttal pdf should be part of the main paper, as this is important to guide future work around your method.
> >
> > Regarding the circular dependence when using embeddings I am not sure if I follow the authors arguments and would encourage them to think about the effects of using a changing embedding function as approximation of some fixed embedding function $f^*$.
> >
> > But overall many of my doubts or concerns have been answered and I definitely think the Bayesian perspective on DPS offers a novelty that makes this work stand out, so I have increased my original score from 6 to 7.

---

> ### Author Response · Authors · 2024-08-13
>
> Regarding the embedding functions: In practice, using a fixed embedding function requires an additional embedding model, which increases memory usage. Additionally, the embedding model may require tuning, which would add extra time. We could consider including experiments to empirically understand the gap between using a changing function approximation and a fixed embedding function.
>
> We sincerely appreciate the reviewer’s insightful and valuable feedback, as well as their time. We will incorporate the information provided in the rebuttal into the revised paper.

---

### Official Review · Reviewer_oLJs · 2024-07-12

**Soundness:** 3
**Presentation:** 3
**Contribution:** 3
**Rating:** 6
**Confidence:** 4

**Summary:**

This work proposes a Bayesian approach to the data point selection task. Specifically, the authors introduce the important weights to each training point and derive the posterior joint probability of the instance-wise weights and network parameters. The parameters and weights are then sampled iteratively based on the stochastic-gradient MCMC technique. Experiments on computer vision and natural language processing have been conducted to show the efficacy of the proposed method.

**Strengths:**

1. This work views the data point selection problem from the perspective of Bayesian theory and introduces the posterior distributions of the instance-wise weights and network parameters.

2. Experimental results demonstrate the efficacy of the proposed method.

**Weaknesses:**

1. I suggest the authors provide a detailed derivative for Eq. 4.

2. Eq. 5 still requires to compute the normalizing constant in Eq. 4, which serves as the bias of the gradient.

3. The authors claim that “Our method straightforwardly achieves sparsity on the w weights allowing efficient sample selection unlike”. I hope the authors provide more analysis and evidence about the sparsity of instance-wise weight.

**Questions:**

Please see the weaknesses

**Limitations:**

No potential negative societal impact.

---

> ### Author Rebuttal · Authors · 2024-08-06
>
> ## [Re: Derivations for Eq 4]
> See [Re-1: Derivations for Eq 4] in the General Rebuttal
>
> ## [Re: Eq5 and Normalizing Constant]
>
> No. The normalising constant in Eq.(4) is $p(D_m|D_t)$ (please see our derivations, especially, Eq.(R4) in our rebuttal above), and this normalising constant has no dependency on $\theta$ and $\lambda$. Thus after taking the gradient with respect to $\theta$ and $\lambda$, the normalising constant simply disappears. This technique is well known for Bayesian posterior sampling [Welling & Teh] and also for sampling from energy-based models, eg, T. Du, and I. Mordatch, "Implicit Generation and Modeling with Energy Based Models," NeurIPS 2019.
>
> ## [Re: Sparsity]
>
> Although we have observed clear distinction in the learned weights between relevant and irrelevant data points (eg. our proof of concept experiments), the weights are not as pronounced as 0/1 like binary selection. We believe that this issue might be diminished by incorporating an additional prior/regularizer that encourages 0/1 weight values. We will investigate it further along this line in our future work.

---

> ### Author Response · Authors · 2024-08-13
>
> Dear reviewer oLJs,
>
> We appreciate your thoughtful and detailed feedback once again.
>
> Since today is the final day for the rebuttal discussion, we wanted to check if you have any remaining questions or suggestions that we can address before the deadline.
>
> Thank you for your time.

---

### Official Review · Reviewer_JeEz · 2024-07-12

**Soundness:** 3
**Presentation:** 3
**Contribution:** 2
**Rating:** 7
**Confidence:** 4

**Summary:**

This paper proposes a new Bayesian method for Data Point Selection (DPS), called BADS. DPS aims to select training data points that optimize performance on downstream tasks. Instead of relying on bi-level optimization (BLO), BADS frames DPS as a posterior inference problem. The method uses a weight network to learn the importance of each data point and proposes a novel update rule for jointly updating model parameters and data point weights using Stochastic Gradient Langevin Dynamic sampling. Experiments on MNIST, CIFAR, and WebNLG datasets show that BADS outperforms or matches existing DPS methods, including BLO, in terms of accuracy and efficiency.

**Strengths:**

- Originality: This paper introduces a novel perspective on DPS by using a Bayesian framework instead of the BLO approach. This offers a new way to think about data selection and its connection to posterior inference. The paper also leverages Langevin sampling for optimizing the derived posterior inference which adds to the novelty.
- Clarity: The paper is well-written and easy to understand. The authors clearly explain their method and the motivations behind their design choices. The final derived method is simple and intuitive.

**Weaknesses:**

- Limited theoretical analysis: The paper focuses on the empirical performance of BADS and lacks a deep theoretical analysis. Providing more theoretical insights on the convergence and optimality properties of BADS would strengthen the paper, especially since the authors claim that their method, unlike BLO, has theoretical convergence guarantees.
- Hyperparameter sensitivity: The paper mentions that BADS has several hyperparameters that require careful tuning. A more in-depth discussion on the impact of these hyperparameters and potential strategies for automatic tuning would be beneficial.
- Comparison and claims:  The paper primarily compares BADS with BLO, but there are other methods that deal with selecting data to learn on, such as offline pruning methods [1], and online and offline batch selection [2, 3], among others. In addition to that, some of these methods have already been used for large scale model training, such as [4, 5].

Some important equations and their corresponding assumptions have been omitted. In particular, equation (4) is not derived and on first glance seems to assume independence of $w$ and $\mathcal{D}$, but it is not clear if this still holds after the introduction of the weight network.

1. https://arxiv.org/abs/2206.14486
2. https://arxiv.org/abs/2312.05328
3. https://arxiv.org/abs/2406.10670
4. https://arxiv.org/abs/2402.09668
5. https://arxiv.org/abs/2406.17711

**Questions:**

Can the authors provide any theoretical guarantees on the convergence of BADS? What are the conditions under which BADS finds an optimal solution?

What is the impact of different hyperparameters, especially the sparsity level parameter and impact constants? How robust is BADS to hyperparameter variability.

**Limitations:**

The authors have adequately addressed the limitations proposed in their work.

---

> ### Author Rebuttal · Authors · 2024-08-04
>
> ## [Re: Convergence Analysis]
>
> We provide a theorem showing that our SGLD algorithm converges to the true posterior to some extent. Our analysis is based on (Zou et al. 2021) where we make some adjustments for our case.
>
> **Assumption 1** (from Assumption 4.3 of (Zou et al. 2021))
> There exists $m>0$ and $b\geq 0$ such that
>
> $$\bigg\langle\nabla_{\theta,w}\log p(\theta,w|D_t,D_m),[\theta, w]\bigg\rangle\geq m ||[\theta, w]||_2^2-b$$
>
> holds for any $\theta,w$.
>
> **Assumption 2** (from Assumption 4.4 of (Zou et al. 2021))
> Any minibatch gradient of the log-posterior is Lipschitz continuous. Ie, there exists a constant $L$ such that for any $z_i^t\in D_t$ and $z_j^m\in D_m$,
>
> $$
> ||\nabla_{\theta,w} A(\theta,w)-\nabla_{\theta',w'} A(\theta',w') ||_2\leq L ||[\theta, w]-[\theta', w']||_2
> $$
> holds for any $\theta,w,\theta',w'$, where $A(\theta,w)=\log p(w)+\log p(\theta)-N_t w_i l(z_i^t;\theta)-N_m l(z_i^m; \theta)$.
>
> Then we have the following convergence theorem -- We omit detailed formulas due to space limit, but can be found in (Zou et al. 2021).
>
> **Theorem** (from Theorem 4.5 of (Zou et al. 2021))
> Let $B$ be the batch size and $\eta$ the step size. Suppose Assumption 1 and 2 hold. For any $\epsilon\in(0,1)$, with the initial iterate satisfying a certain ball constraint, the distribution $\mu_K^{SGLD}$ of the $K$-th iterate in our SGLD iterations Eq.(6-7) satisfies:
>
> $$||\mu_K^{SGLD}-p(\theta,w|D_t,D_m)||_{TV} \leq C(B,\eta,K,\epsilon)+\epsilon/2$$
>
> where $C(B,\eta,K,\epsilon)$ is constant that can be reduced by adjusting $B$, $\eta$ and $K$, and $||\cdot||_{TV}$ stands for the total variation distance.
>
> (Zou et al. 2021) "Faster Convergence of Stochastic Gradient Langevin Dynamics for Non-Log-Concave Sampling", UAI 2021.
>
> ## [Re: Hyperparameter Analysis]
>
> See [Re-3: Hyperparameter Analysis] in the General Rebuttal
>
> ## [Re: Comparison and Claims]
>
> We thank the reviewer for bringing the recent related work to our attention. We recognize that there are three types of DPS setups based on supervision signals:
>
> * **Unsupervised DPS**: DPS **without** the guidance of a held-out meta set involves models selecting a subset from a training set based on specific hypotheses, such as "challenging examples improve model performance". Curriculum learning falls into this category. This method is also widely used in pre-training data point selection (**[1]** and **[4]**).
>
> * **Self-supervised DPS**: DPS **with** a held-out meta set. However, the meta set **does not share the same data distribution** as the targeted test set. Typically, the examples in the meta set are also selected from the training set based on specific hypotheses, such as "learnable examples enhance model performance". **[2]**, **[3]** and **[5]** fall into this category.
>
> * **Meta-set guided DPS**: DPS **with** a small meta set that **shares the same distribution** as the test set, in order to train a model that performs well specifically on the target test set. The test set may encompass one or multiple downstream domains or tasks. This DPS is closely related to meta learning, domain adaptation, and transfer learning. Existing approaches are predominantly based on **BLO**. **Our work**, as explained in Section 2.1, focuses on this type.
>
> To broaden our comparison, we choose studies **[4]** and **[2]** from the first two categories above respectively and evaluate them using our experimental setups. Here are the results:
>
> *  **Compared to Self-supervised DPS**:
>     * To guide DPS, **[2] (ClassAct)** investigates using a reference model trained on a meta set chosen from the training set based on predefined learnability metrics. To fairly compare the selection mechanism, we replaced their meta set using our meta set.
>     * **Figure 4 in the PDF** shows that ClassAct performs worse than BADS with a large margin across all three proof-of-concept scenarios. It completely failed in the Data Balancing and Efficient Learning scenarios.
>     * **Figure 5 in the PDF** shows that the weights predicted by ClassAct for the training examples are unreliable.
>     * **Table 1** below shows that BADS outperform ClassAct on all LLM benchmarks.
>     * **Table 2** below shows that BADS requires significantly less GPU memory and computing time compared to ClassAct.
>
> *  **Compared to unsupervised DPS**:
>     * Paper **[4] (AskLLM)** selects examples from training set by prompting LLMs. We compare to it only in the LLM use case due to two reasons:
>         * Paper **[1]** already showed that unsupervised DPS can amplify class imbalances.
>         * Open-source LLMs typically do not handle vision input.
>     * For fair comparison, we compare BADS with an online variant of [4] (dubbed **AskLLM-O**) where we used the pretrained OpenLLaMA 3B to obtain the sampling score for each training sample.
>     * **Table 1** below shows that BADS outperforms AskLLM-O on 2 out of 4 LLM benchmarks.
>     * **Table 2** below shows that BADS requires significantly less compute and memory compared to AskLLM-O. AskLLM-O also requires an offline scoring of all training samples which took 13932 seconds (3.87 hours) in our setup on a single NVIDIA A40 GPU.
>
> **[Table 1]** DPS performance in LLM use case
> |Models | MMLU  | ARCc | ARCe | HellaSwag |
> |-----| ------------- | ------------- |  ------------- | ------------- |
> | BADS |  26.59 | 34.39 | 67.00 | 52.91 |
> | ClassAct | 24.90 | 31.91 | 67.34 | 52.09 |
> | AskLLM-O | 25.55 | 35.15 | 66.88 | 53.71 |
>
> **[Table 2]** DPS memory and time usage over 100 steps
> |Models | Avg-Memory (MB) | Time (s) |
> |-----| ------------- | ------------- |
> | BADS | 14694.58 | 61.03 |
> | ClassAct | 34821.36 | 269.81 |
> | AskLLM-O | 32908.89 | 115.24 (+13932) |
>
> We also note that:
> * An online variant of [3] is similar to the CDS baseline in our paper.
> * [3] and [5] were released after the Neurips deadline
>
> ## [Re: Graphical Model and Derivations]
> See the General Rebuttal:
> * [Re-1: Derivations for Eq 4]
> * [Re-2: Graphical Model and Derivations with Weight NN]

---

> > ### Comment · Reviewer_JeEz · 2024-08-12
> >
> > I would like to thank the authors for thoroughly addressing my comments and quickly incorporating the revelant comparisons to related work and baseline comparisons where relevant. This new information has improved my understanding and confidence in the impact of the paper.

---

> ### Author Response · Authors · 2024-08-13
>
> We sincerely appreciate the reviewer’s insightful and valuable feedback, as well as their time. We will incorporate the information provided in the rebuttal into the revised paper.

---

### Author Rebuttal · Authors · 2024-08-05

We sincerely thank all reviewers for their thoughtful and detailed feedbacks.

## [Re-1: Derivations for Eq 4] @Reviewer **JeEz**, **oLJs**, and **5MKv**
From the graphical model in **Fig. 1 in PDF**, $D_m \perp (w, D_t) \ | \  \theta$ is the only conditional independence assumption we make.
$$
p(\theta,w|D_t,D_m)=\frac{p(\theta,w,D_m|D_t)}{p(D_m|D_t)}
$$
$$
\ \ =\frac{1}{p(D_m|D_t)}\cdot p(w)\cdot p(\theta,D_m|w,D_t)
$$
$$
\ \ =\frac{1}{p(D_m|D_t)}\cdot p(w)\cdot p(D_m|\theta,w,D_t)\cdot p(\theta|w,D_t)
$$
$$
\ \ = \frac{1}{p(D_m|D_t)}\cdot p(w)\cdot p(D_m|\theta) \cdot p(\theta|w,D_t) \ \ (R1)
$$
$$
\ \ \propto\cdot p(w)\cdot p(D_m|\theta)\cdot p(\theta|w,D_t) \ \ (R2)
$$
where in (R1), we use $D_m \perp (w, D_t) \ | \  \theta$, and in (R2) $\frac{1}{p(D_m|D_t)}$  is regarded as constant for it has nothing to do with $\theta$ and $w$.

## [Re-2: Derivations with Weight NN] @Reviewer **JeEz**, **oLJs**, and **5MKv**
Here are the full details of possible changes when the weight network is adopted.

First, the weights $w$ become a deterministic function of $D_t$ and $\phi$ (here $\phi=$ weightnet params) as shown in Fig. 1(a) in the attached PDF.  But since $w$ is a deterministic function of $\phi$ and $D_t$, we can simplify it by having $w$ absorbed into $\phi$ while introducing conditional dependence (an arrow) from $D_t$ to $\phi$, as depicted in Fig. 1(b) in the PDF.

Note that $D_t$ is always given, and we treat $\phi$ as random variables, and wherever $w_i$ appears, we replace $w_i$ by $weightnet(z_i^t; \phi)$. More specifically, we make the following changes to the equations in the weight net scenario:

Eq.(2): $p(\theta|\phi,D_t) \propto p(\theta) \cdot \prod_{i=1}^{N_t} p(\textrm{weightnet}(z_i^t; \phi), z^i_t | \theta)$

Eq.(8): $p(\phi|D_t) \propto e^{-(\sum_i weightnet(z_i^t;\phi)-\lfloor N_t\beta\rfloor)^2 / 2\sigma^2}$

Eq.(4) (detailed derivations):

$$
p(\theta,\phi|D_t,D_m) = \frac{p(\theta,\phi,D_m|D_t)}{p(D_m|D_t)}
$$
$$
\ \ \ \ = \frac{1}{p(D_m|D_t)} \cdot p(\phi|D_t) \cdot p(\theta,D_m|\phi,D_t)
$$
$$
\ \ \ \ = \frac{1}{p(D_m|D_t)} \cdot p(\phi|D_t) \cdot p(D_m|\theta,\phi,D_t) \cdot p(\theta|\phi,D_t)
$$
$$
\ \ \ \ = \frac{1}{p(D_m|D_t)} \cdot p(\phi|D_t) \cdot p(D_m|\theta) \cdot p(\theta|\phi,D_t)
$$
$$
\ \ \ \ \propto \cdot p(\phi|D_t) \cdot p(D_m|\theta) \cdot p(\theta|\phi,D_t)
$$

Eq.(5): $[\theta, \phi] \ \leftarrow \ [\theta, \phi] + \frac{\eta}{2} \nabla_{\theta,\phi} \log p(\theta,\phi|\mathcal{D}_t,\mathcal{D}_m)  + \epsilon \sqrt{\eta}, \ \ \ \ \epsilon\sim\mathcal{N}(0,I)$

Eq.(6): $\theta \ \leftarrow \ \theta + \frac{\eta}{2} \nabla_{\theta} ( \log p(\theta) - N_t \cdot E_{i\sim B_t}[weightnet(z_i^t;\phi) \cdot l(z_i^t;\theta)] - N_m\cdot E_{j\sim B_m}[l(z_j^m;\theta)] ) + \epsilon_\theta \sqrt{\eta}$

Eq.(7): $\phi \ \leftarrow \ \phi + \frac{\eta}{2} \nabla_{\phi} ( \log p(\phi|D_t) - N_t\cdot E_{i\sim B_t}[weightnet(z_i^t;\phi) \cdot l(z_i^t;\theta)] ) + \epsilon_\phi \sqrt{\eta}$


## [Re-3: Hyperparameter Analysis] @Reviewer **JeEz** and **5MKv**

We will incorporate a detailed discussion and ablation study on this topic in the revised paper. Here is a summary:

**Two main sets of hyperparameters for BADS**:
* For parameters update: $\eta$, $\epsilon_\theta$, $ \epsilon_w$, $\rho_{\theta }^{t}$, $\rho_{\theta }^{m}$, and $\rho_{w}^{t}$ (*Eq 6, 7 and Appendix B*).
* For the prior distribution of the example weights: $\sigma$, $\beta$, $s_{avg}$ (*Eq 8,9*) .

**Hparams tuning principles**:
* We set $\eta$ to 1 and kept the Gaussian noise small with $\epsilon_\theta$ and $\epsilon_w$ equal to 1e-5.
* $\rho_{w}^{t}$ is set to 1.
* We set $\rho_{\theta }^{m}$ to 1 and primarily adjust $\rho_{\theta }^{t}$. In most cases, $\rho_{\theta }^{t}$ is simply set to 1. However, if the training set contains noise—where the ground truth labels might be incorrect—the loss from the training examples, particularly in the early stages of training, may be unreliable. In such cases, we decrease $\rho_{\theta }^{t}$.
* $s_{avg}$ should not be too large, as it may incorporate outdated weights from earlier training steps. We set it to 10.
* Generally, we select $\beta$ based on the proportion of training data we consider beneficial for the downstream tasks. For LLMs, we adopt the same ratio used in the previous studies (*paper [52] and [46]*).
* $\sigma$ controls how tightly the weights should match $\beta$. We set $\sigma$ based on our confidence in the selection ratio $\beta$: a smaller $\sigma$ indicates greater confidence in $\beta$.

Data Denoising scenario is a bit special, Eq 7 shows that high losses from the training batch push the example weights $w$ toward 0. With a high noise rate (50% and 80%), the training batch losses remain high throughout the training process. To prevent the weights $w$ from collapsing to 0, we use a high selection ratio $\beta$ and a low $\sigma$.

**Ablation study conclusions: Influence of the impact constants $\sigma$ and sparsity level $\beta$**:
* Higher $\sigma$ causes the example weights $w$ to drift away from $\beta$, occasionally collapsing to 0, and may result in incorrect example weights (See **Figure 2 in PDF**). In all three proof-of-concept scenarios, the models achieve similarly good performance when $\sigma$ is reduced to around $10^{-5}$.
* In both WebNLG and MNIST scenarios, high $\beta$ leads to incorrect example weights (See **Figure 3 in PDF**). Conversely, due to the reason we mentioned above, in data denoising (CIFAR) scenario, lower $\beta$ leads to incorrect weights.
* In both the WebNLG and MNIST scenarios, the backbone models' performance significantly declines when the example weights are incorrect (See **Figure 1 in PDF**). In the CIFAR scenario, the impact is less pronounced.

Due to space constraints, we present a selective set of plots. The complete results for the three proof-of-concept scenarios will be provided in the revised paper.

---

### Decision · Program_Chairs · 2024-09-25

**Decision:**

Accept (poster)

**Comment:**

All reviewers unanimously voted for acceptance. Weaknesses like missing baselines in the experiments and missing theoretical results, were outbalanced during the rebuttal period. Authors are strongly encouraged to import the additional results into the paper.